**Data Availability Statement:** The ethics committee of the Zahedan University of Medical Sciences,

# Potential efficacy of caffeine ingestion on balance and mobility in patients with multiple sclerosis: Preliminary evidence from a single-arm pilot clinical trial

**Afsoon Dadvar**[1], **Melika Jameie** [2,3]*, **Mehdi Azizmohammad Looha**[4], **Mohammadamin Parsaei** [5], **Meysam Zeynali Bujani**[1], **Mobina Amanollahi**[5], **Mahsa Babaei**[6], **Alireza Khosravi**[7], **Hamed Amirifard** [2]*

**1** Student Research Committee, Zahedan University of Medical Sciences, Zahedan, Iran, **2** Iranian Center of Neurological Research, Neuroscience Institute, Tehran University of Medical Sciences, Tehran, Iran, **3** Neuroscience Research Center, Iran University of Medical Sciences, Tehran, Iran, **4** Basic and Molecular Epidemiology of Gastrointestinal Disorders Research Center, Research Institute for Gastroenterology and Liver Diseases, Shahid Beheshti University of Medical Sciences, Tehran, Iran, **5** School of Medicine, Tehran University of Medical Sciences, Tehran, Iran, **6** Headache Department, Iranian Center of Neurological Research, Neuroscience Institute, Tehran University of Medical Sciences, Tehran, Iran, **7** Clinical Immunology Research Centre, Department of Neurology, School of Medicine, Zahedan University of Medical Sciences, Zahedan, Iran

☯ These authors contributed equally to this work.

* Dr.amirifard@gmail.com (HA); Jameiemelika@gmail.com, Ms-jameie@farabi.tums.ac.ir, Jameiemelika@sbmu.ac.ir (MJ)

## Abstract

### Objectives

Caffeine's potential benefits on multiple sclerosis (MS), as well as on the ambulatory performance of non-MS populations, prompted us to evaluate its potential effects on balance, mobility, and health-related quality of life (HR-QoL) of persons with MS (PwMS).

### Methods

This single-arm pilot clinical trial consisted of a 2-week placebo run-in and a 12-week caffeine treatment (200 mg/day) stage. The changes in outcome measures during the study period (weeks 0, 2, 4, 8, and 12) were evaluated using the Generalized Estimation Equation (GEE). The outcome measures were the 12-item Multiple Sclerosis Walking Scale (MSWS-12) for self-reported ambulatory disability, Berg Balance Scale (BBS) for static and dynamic balance, Timed Up and Go (TUG) for dynamic balance and functional mobility, Multiple Sclerosis Impact Scale (MSIS-29) for patient's perspective on MS-related QoL (MS-QoL), and Patients' Global Impression of Change (PGIC) for subjective assessment of treatment efficacy. GEE was also used to evaluate age and sex effect on the outcome measures over time. (Iranian Registry of Clinical Trials, IRCT2017012332142N1).

### Results

Thirty PwMS were included (age: 38.89 ± 9.85, female: 76.7%). Daily caffeine consumption significantly improved the objective measures of balance and functional mobility (BBS; P-

**Funding:** The authors received no specific funding for this work.

**Competing interests:** The authors have declared that no competing interests exist.

value<0.001, and TUG; P-value = 0.002) at each study time point, and the subjective measure of MS-related QoL (MSIS-29; P-value = 0.005) two weeks after the intervention. Subjective measures of ambulatory disability (MSWS-12) and treatment efficacy (PGIC) did not significantly change. The effect of age and sex on the outcome measures were also assessed; significant sex-time interaction effects were found for MSWS-12 (P-value = 0.001) and PGIC (P-value<0.001). The impact of age on BBS scores increased as time progressed (P-value = 0.006).

## Conclusions

Caffeine may enhance balance, functional mobility, and QoL in PwMS. Being male was associated with a sharper increase in self-reported ambulatory disability over time. The effects of aging on balance get more pronounced over time.

## Trial registration

This study was registered with the Iranian Registry of Clinical Trials (Registration number: IRCT2017012332142N1), a Primary Registry in the WHO Registry Network.

## 1. Introduction

Around the world, 2.8 million people are living with multiple sclerosis (MS) [1]. The majority of persons with MS (PwMS) experience balance and gait dysfunction (ambulatory disabilities) even early in the disease course [2]. A variety of MS-related symptoms (e.g., weakness, spasticity, fatigue, and coordination alterations) can affect balance, postural control, gait, and risk of falling in PwMS [3]. Ambulatory dysfunction can remarkably influence patients' quality of life by increasing personal dependency, limiting daily activities (going to the bathroom on time, crossing the street safely, shopping properly, and working efficiently), instilling fear of falling, and falling-related injuries [2].

Various approaches have been utilized to improve different aspects of functioning, including cognitive and motor function, in patients with neurological diseases [4,5]. Interventions, including physical rehabilitation [6], exercise, non-invasive brain stimulation [7,8], and medications such as dalfampridine (4-aminopyridine) [9–11], nabiximols [12], polyunsaturated fatty acids, omega-3, omega-6 [13], and lipoic acid [14], have been evaluated for balance and gait improvement in PwMS [2,6–15]. Notably, dalfampridine is the only U.S. Food and Drug Administration (FDA)-approved medication for improving the balance and walking abilities of PwMS (Ampyra (dalfampridine) Information | FDA) [9–11,15]. However, it should be prescribed with caution, as it may cause serious side effects [16], including severe allergic reactions, seizures, and triggering/exacerbating trigeminal neuralgia (medication guide available at label (fda.gov)). Additionally, it seems that dalfampridine may help only a subset of PwMS, with one-quarter and one-third of patients experiencing faster walking speed and enhanced walking ability, respectively [16].

Caffeine, a natural compound, is the most widely consumed psychoactive agent in the world [17]. Studies have shown caffeine's potential benefits on various neurological disorders, including seizure, Alzheimer's disease, Parkinson's disease, stroke, and MS [18–25], possibly by reducing neuroinflammation and oxidative stress and increasing neurogenesis [26,27]. Specifically, research has highlighted the positive effects of caffeine on the ambulatory performance of non-MS populations [28–30], as well as on selected aspects of MS, including

attention and disease progression [18–24]. Aligned with caffeine's impact on the central nervous system, skeletal muscles [31], the ambulatory performance of non-MS populations [28–30], and specific aspects of MS [18–20,23,24], potentially favorable effects on balance and mobility in PwMS could also be anticipated. Consequently, we hypothesized that caffeine might have the potential to improve balance and mobility impairments as debilitating aspects of MS.

Despite supportive evidence [18–20,23,24,28–30], to our knowledge, the potential effect of caffeine on balance and mobility in PwMS has not yet been studied. In this pilot single-arm phase II clinical trial we evaluated the potential effectiveness of caffeine ingestion on balance and mobility (i.e., static and dynamic balance, functional mobility, and patient's reported ambulatory disability) among PwMS. Additionally, we investigated caffeine's potential effect on patients' health-related quality of life (HR-QoL).

## 2. Materials and methods

### 2.1. Trial design and any changes after trial commencement

This was a pilot single-arm phase II clinical trial. Notably, the initial study protocol involved a double-armed design, with one group receiving caffeine and another group receiving a placebo. However, due to financial constraints encountered during the study, it became necessary to modify the protocol and proceed as a single-armed study. To mitigate potential bias, a two-week placebo run-in stage was incorporated before the investigation. Furthermore, although the initial protocol was designed for a caffeine dosage of 2.5 mg/kg/day, we were compelled to utilize available caffeine tablets (200 mg/tablet) in the trial due to practical constraints and the unavailability of an oral caffeine solution. The selection of a 200 mg dosage was rationalized on two grounds: Firstly, 200 mg of caffeine corresponds to 2.5–3 mg/kg for an adult weighing between 60–70 kg [31]. Secondly, according to the European Food Safety Authority (EFSA), caffeine doses of up to 200 mg do not elicit safety concerns in non-pregnant adults [32]. Deviations from the protocol were implemented without adversely affecting the rights and well-being of the participants, and all changes were made in consultation with the university's IRB to ensure the upholding of ethical standards.

### 2.2. Participants

Patients who were referred to our specialized MS clinic during the study period were assessed for eligibility. Men and women who met the following criteria were included: (a) aged between 20–55 years, (b) weight > 40 kg, (c) diagnosed with MS according to the McDonald criteria [33], which were confirmed by an expert neurologist regardless of the disease course, (d) having the ability to stand upright for $\geq$ 180 seconds without any support and to still ambulant, with an Expanded Disability Status Scale (EDSS) < 6.0, (e) without clinical relapse/disease progression in the previous three months, and (f) without coexisting conditions or with stable and well-controlled coexisting conditions (stable condition was defined as no change in the types and dosage of medication or disease severity over 3 months before the enrolment). EDSS is the most common tool for measuring the disability level in PwMS, scoring from 0 to 10, with higher scores indicating a higher disability level [34]. Patients with an EDSS score < 6 can walk without aid for at least 100 meters [34].

Main exclusion criteria included: (a) pregnancy or lactation, (b) MS relapse, corticosteroid treatment, or disease progression three months before the study, during the investigation, or the follow-ups, (c) other neurological, psychiatric, or systemic disorders affecting motor function (i.e., seizure, tremor, insomnia, depression with at least moderate severity assessed by Beck Depression Inventory-II (BDI-II) [35], moderate or severe anxiety assessed by The

Persian-validated translation of Hospital Anxiety and Depression Scale-Anxiety subscale (HADS-A$\geq$ 11) [36–38], significant cardiovascular disorders, orthopedic problems, respiratory failure, myopathy, or vestibular disorders (any evidence of peripheral vestibulopathy determined through cerebellar examinations, nystagmus examination, and head impulse test), (c) significant cognitive impairment (defined as taking more than 90 seconds to complete the Trail-Making Test-B) [39–41], (d) medical therapy alterations in the previous 3 months or during the investigation, (e) using concurrent medications that can affect balance (i.e., selective serotonin reuptake inhibitors [SSRIs] and serotonin-norepinephrine reuptake inhibitors [SNRIs] [42]), (f) use of medications with major interactions with caffeine (i.e., dipyridamole and isocarboxazid, linezolid, etc. [43]), (g) hypersensitivity to caffeine, herbal extracts or dietary supplements, and (h) history of hypertension, cardiovascular disease, migraine headaches, renal or liver impairments, peptic ulcer disease, and drug/alcohol abuse [43]. Eventually, (i) patients with an inability or lack of interest to comply with the study procedures (taking medications, responding to phone calls, attending the clinic for revisits) were excluded.

### 2.3. Study settings and ethics statement

This study was conducted at an academic hospital complex affiliated with Zahedan University of Medical Sciences, Zahedan, Iran, and followed the Consolidated Standards of Reporting Trials (CONSORT) extension to pilot and feasibility trials [44] (**S1 Table in S1 File**). The trial was approved by the university's Institutional Review Board (IRB, IR.ZAUMS.REC.1395.236) and registered with the Iranian Registry of Clinical Trials (IRCT) (IRCT2017012332142N1). The first patient received treatment on March 9, 2017, and the trial ended on January 2, 2018. **Fig 1** illustrates the study design and schedule. Patients' eligibility was assessed during a screening visit (visit 0), and those who qualified scheduled a follow-up visit a week later (visit 1) to receive placebo medication for 2 weeks (the 2-week pre-intervention placebo run-in stage). After the placebo run-in stage, patients started taking experimental caffeine tablets (200 m/day) (visit 2, week 0) for a 12-week period, during which four in-person follow-up visits (visits 3–6) were scheduled (the 12-week post-intervention treatment stage). During the study period, patients were instructed to stop consuming caffeinated products, such as coffee, tea, cola drinks, and cocoa. Patients were allowed to continue using concomitant MS therapies according to their neurologist's prescription.

Patients' anonymity was protected, and verbal and written informed consents were properly obtained from participants for participation and publication under the Declaration of Helsinki [45] and Good Clinical Practice [46]. The research team had access to identifiable participant data during the data collection phase. However, steps were taken to ensure that any identifying information was securely stored and separated from the main dataset. After data collection, all personally identifiable information was removed and replaced with unique identifiers. This anonymized dataset was used for analysis and reporting. The research team adhered to strict protocols to protect participant privacy and complied with all applicable data protection regulations. Patients were given the option to drop out of the study, and if they chose to continue, giving an informed re-consent was necessary. An independent neurology specialist performed and scored the tests. All tests were performed by the same neurologist during the first visit, as well as in each follow-up revisit.

### 2.4. Intervention

A 200 mg/day orally administered caffeine tablet (Karen Pharma & Food Supplement Co.Ⓡ) was assessed for its potential efficacy on balance, mobility, and MS-related QoL in PwMS over 12 weeks.

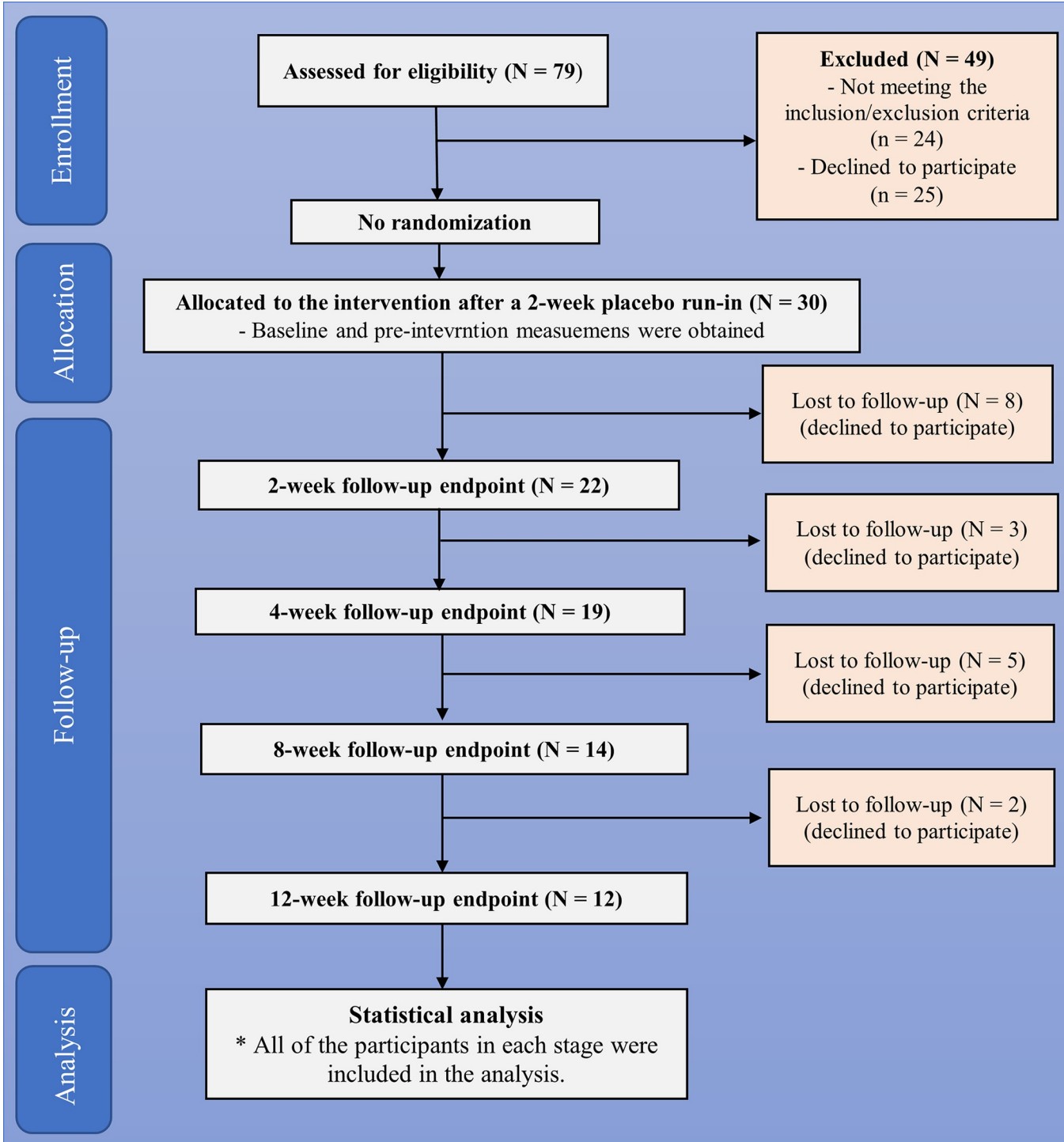

**Fig 1. Study schedule and design.** After a screening visit (visit 0) for eligibility evaluation, eligible patients returned 1 week later (visit 1) and received placebo medication for 2 weeks (the 2-week placebo run-in stage). At visit 2, patients started taking a 200 mg daily dosage of caffeine tablet and returned for four follow-up visits (visits 3–6) over a 12-week period. Abbreviations: MSWS-12: 12-item Multiple Sclerosis Walking Scale, BBS: Berg Balance Scale, TUG: Timed Up-and-Go, MSIS-29: Multiple Sclerosis Impact Scale, PGIC: Patients' Global Impression of Change.

## 2.5. Outcomes

The primary objective of this study was to determine the potential efficacy of an experimental 200 mg/day orally available caffeine tablet (Karen Pharma & Food Supplement Co.®) on balance, mobility, and MS-related QoL of PwMS. The outcome measures included (a) the 12-item Multiple Sclerosis Walking Scale (MSWS-12) for patient's reported ambulatory disability [47], (b) the Berg Balance Scale (BBS) for static and dynamic balance [48], (c) the Timed Up-and-Go (TUG) for functional mobility and dynamic balance [49], (d) the Multiple Sclerosis Impact Scale (MSIS-29) for patients' perspective on MS impact on their QoL [50], and (e) the Patients' Global Impression of Changes (PGIC) to evaluate patients' own opinions on the treatment efficacy [51]. The outcome measures were assessed at baseline and after 2, 4, 8, and 12 weeks.

As the secondary objectives, we also examined the effects of sex and age on patients' balance, mobility, and MS-related QoL over time. Finally, the correlations between balance and mobility-related measures (MSWS-12, BBS, and TUG), MS-related QoL (MSIS-29), and the patients' opinions on treatment efficacy (PGIC) were studied.

## 2.6. Study measures

Patients' baseline characteristics (age, sex, past medical history, habitual history, education, and job status), MS-related characteristics (MS phenotype, age of disease onset, disease duration, EDSS, and using disease-modifying drugs [DMDs]), balance and mobility-related measures (MSWS-12, BBS, TUG), and QoL measure (MSIS-29) were documented in week 0 (visit 2). During the following three post-intervention months, MSWS-12, BBS, TUG, MSIS-29, and PGIC were reassessed in each follow-up visit at week 2 (visit 3), week 4 (visit 4), week 8 (visit 5), and week 12 (visit 6) (**Fig 1**). Herein, we briefly describe the outcome measures:

**The 12-item Multiple Sclerosis Walking Scale (MSWS-12):** a self-report questionnaire with good test-retest reliability [52], consisting of 12 items that indicate the patient's reported ambulatory disability caused by MS [47]. Each question earns a score of 1–5, with a total score ranging from 12–60. Higher scores indicate poorer performance [47] (**S2 Table in S1 File**). A valid and reliable Persian translation was used in this study [53].

**The Berg Balance Scale (BBS):** an objective 14-item scale, ranging from 0–56, which is used to determine the patient's ability to safely balance during a series of predetermined tasks [48]. BBS assesses both static and dynamic balance with high reliability [48,54,55]. Each test item consists of a five-point ordinal scale ranging from zero to four, with zero indicating the lowest and four indicating the highest level of function [48] (**S3 Table in S1 File**). A validated Persian translation of the test was used in this study [56].

**The Timed Up-and-Go (TUG):** an objective test with high test-retest reliability [52], measuring dynamic balance and functional mobility [49]. The test requires that a patient stands from a chair, walks a distance of 3 meters, 180° turns around a cone, and then returns to a seated position as quickly as possible (TUG_test-print.pdf (cdc.gov); **S1 Fig in S1 File**) [49]. The faster the TUG speed, the better the mobility and dynamic balance [49].

**The Multiple Sclerosis Impact Scale (MSIS-29):** a self-report disease-specific health-related QoL (HR-QoL) measure with a high test-retest reliability [50], explaining the patient's perspective on the disease's impact on their QoL through the last two weeks. This 29-item questionnaire consists of 20 items related to the physical scale and 9 items related to the psychological scale [50]. Each item has five response options from one (not at all; the best) to five (extremely; the worst) (**S4 Table in S1 File**).

**The Patient's Global Impression of Change (PGIC):** a seven-point self-report scale evaluating the patient's overall health status changes after taking a specific treatment course [51]. This

scale ranges from one to seven, with higher scores indicating a more considerable improvement (Official PGI-C, PGI-I, PGI-S distributed by Mapi Research Trust | ePROVIDE (mapi-trust.org)) [51] (**S5 Table in S1 File**).

## 2.7. Sample size

According to the sample size rule of thumb for pilot trials suggested by Browne *et al.*, a sample size of 30 patients was initially calculated [57,58]. Additionally, the determination of sample size for this study involved the utilization of the GPower software. A deliberate underestimation was integrated, employing a conservative approach with an effect size established at 0.25, intentionally lower than the actual expected difference and effect size in our investigation. The analysis was conducted with a significance level of 0.05, and a range of power values from 0.8 to 0.95 was examined, incorporating considerations for five measurements and an assumed correlation of 0.5 between repeated measurements, which underestimated the true effect size and correlation observed in the study. Despite the conservative nature of these choices, the resulting sample size was computed as a minimum of 21 for a power of 0.8 and 30 for a power of 0.95.

## 2.8. Randomization

Not applicable.

## 2.9. Blinding

We described the nature of the study and the possibility of receiving either the active drug or a placebo without revealing the specific timing or details of when each would be administered. Participants were informed that at no point during the trial would they know which treatment they were getting, ensuring blinding during each stage, including both the 2-week pre-intervention placebo run-in stage and the 12-week post-intervention treatment stage. To reduce bias, outcome assessors (neurology specialists) were also blinded to the research hypothesis and medication codes. Both caffeine and placebo tablets were similar (in terms of materials, shape, color, taste, and fragrance), and they were packaged in the same bottle type to enable blinding.

## 2.10. Adherence monitoring

Notably, throughout the course of the 14-week trial, medication compliance and avoidance of caffeinated products were carefully assessed, using several approaches [59,60]. First, before obtaining informed consent, patients were educated about the study's objectives, the significance of adhering to study requirements, the need to strictly follow the protocol, and detailed guidance on avoiding caffeinated products. Second, trained study personnel dispensed a certain amount of placebo or active medication during scheduled in-person visits (visits 1–6), ensuring that patients received only the required amount until their next visit. Additionally, at each visit, patients were reminded to bring the medication package from their previous visit and advised to abstain from caffeine-containing items. Third, paper-based patient diaries were employed to meticulously document all information related to medication intake, adverse events, and caffeine product consumption. Fourth, between visits, patients were contacted daily to confirm medication compliance, monitor potential adverse events, evaluate caffeine consumption, and address any other concerns. Eventually, participants were encouraged to reach out to the study team in case of difficulties, side effects affecting their compliance, or any violations involving the consumption of caffeinated products.

## 2.11. Safety monitoring

Regarding safety concerns, available evidence suggests that caffeine consumption is not only not associated with an increased risk of MS development or exacerbation, but it may also potentially reduce susceptibility to MS or ameliorate the disease progression [21,61]. An independent safety monitoring board comprised of three expert neurologists was responsible for assessing any treatment-related adverse events (trAEs). To classify an adverse event as treatment-related, the neurologists took into account the logical temporal association with the treatment administration, the expected patterns of response, and the exclusion of other factors, according to the NIA Adverse Event and Serious Adverse Event Guidelines (available online at NIA Adverse Event and Serious Adverse Event Guidelines (nih.gov)). Solicited trAEs evaluated in this study included CNS symptoms (headaches, lightheadedness, anxiety or agitation, tremor, restlessness, insomnia, seizure, etc.), cardiovascular symptoms (tachycardia, dysrhythmia, hypotension), and gastrointestinal symptoms (nausea, vomiting, abdominal cramping, dyspepsia, diarrhea, anorexia, etc.) [43].

## 2.12. Statistical analysis

Descriptive statistics were presented using mean ± standard deviation (SD) for numeric variables and frequency (%) for categorical variables. Wilcoxon test with Bonferroni adjustment was used to compare changes in scores between pairs of time using bar plots. The generalized estimation equation (GEE) was performed to examine the change in scores during the study period. In addition, GEE was used to evaluate the impact of age and sex on the scores at different times. Analyses were performed using SPSS (version 26) and R (version 4.2.1). Significant results were determined by P-values less than 0.05.

## 3. Results

Fig 2 indicates the CONSORT flow diagram of participants. A total of 79 patients were screened for eligibility, of whom 49 individuals were excluded. Finally, 30 PwMS were included for further evaluation. Within the three-month post-intervention period, at each study time point, eight, three, five, and two patients dropped out of the study due to a lack of willingness. No serious trAEs occurred during the study period.

### 3.1. Descriptive statistics of demographic and MS-related variables

Table 1 shows the demographics and MS-related characteristics of patients. A total of 30 PwMS with a mean EDSS of 4.2 ± 1.3 were evaluated (mean age: 38.89 ± 9.85, female: 76.7%). Most of the patients (76.7%) had a relapsing-remitting course of the disease and were receiving DMD (86.7%). The mean age at MS diagnosis was 29.73 ± 8.82, with a mean disease duration of 8.57 ± 7.13 years.

### 3.2. The trend of criterion scores

Table 2 shows the changes in scores over time for each criterion. The MSWS-12 did not significantly change over time (P-value = 0.583). Nevertheless, the BBS score significantly increased (P-value<0.001), and the time to complete the TUG task significantly decreased (P-value = 0.002) over time. A significant decline was observed in the MSIS-29 score (P-value = 0.005), especially two weeks after the intervention.

Fig 3 illustrates the changes in MSWS-12, BBS, TUG, MSIS-29, and PGIC scores. It was evident that TUG was trending downward in a significant and nearly constant manner. The BBS scores trended in the opposite direction. Furthermore, the MSIS-29 score showed a significant

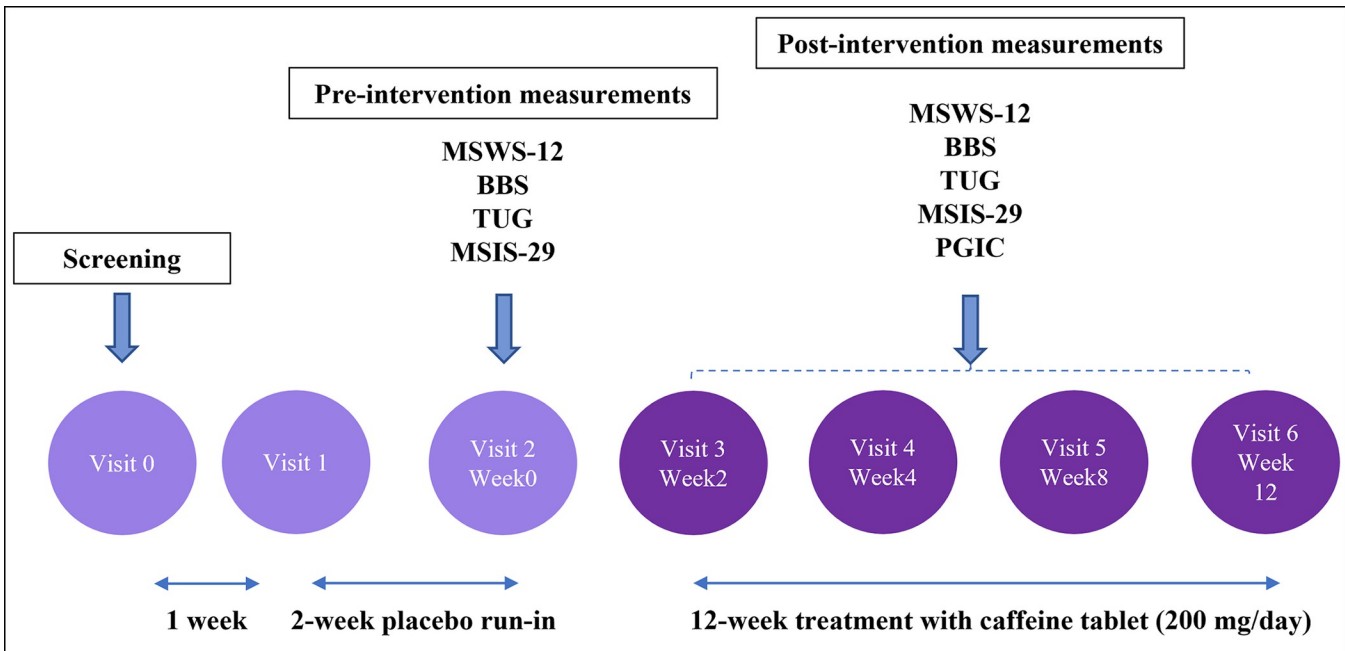

**Fig 2. The CONSORT Flow diagram of the participant disposition.**

decline 2 weeks after the intervention (P-value = 0.005). However, MSWS-12 and PGIC did not show significant changes over time (P-value = 0.583 and 0.213).

### 3.3. Impact of sex on the scores

In the next step, as shown in **Table 3** and **S2 Fig in S1 File,** the impact of sex on scores over time was examined. Accordingly, PGIC was significantly lower for males than for females at the end of the second week (β: -3.12, 95% CI: -4.42, -1.83, P-value<0.001). The interaction effect of sex and time on both the MSWS-12 and PGIC was significant (**S2 Fig in S1 File**). This effect resulted in a significant increase in the MSWS-12 score for males compared to females for every unit increase in time (β: 7.55, 95% CI: 2.95, 12.16, P-value = 0.001). Also, the PGIC slope for males was higher than that for females over time (β: 0.92, 95% CI: 0.66, 1.19, P-value<0.001). No significant interaction effect for sex and time was seen on BBS (P-value = 0.940), TUG (P-value = 0.351), and MSIS-29 (P-value = 0.955) measures.

### 3.4. Impact of age on the scores

**Table 4** shows the impact of age on scores over time. Older age was associated with higher scores of MSIS-29 (β: 1.80, 95% CI: 0.88, 2.73, P-value<0.001) and lower scores of BBS (β: -0.39, 95% CI: -0.77, -0.02, P-value = 0.039) at baseline. The impact of age on the BBS increased significantly for every unit increase in time (β: 0.09, 95% CI: 0.03, 0.15, P-value = 0.006).

### 3.5. The Spearman correlation coefficient

Eventually, we assessed the correlations between outcome measures, including the MSWS-12, BBS, TUG, MSIS-29, and PGIC (**Fig 4**). The results showed significant positive correlations between MSIS-29 and MSWS-12 (r = 0.82, P-value<0.001), MSIS-29 with TUG (r = 0.73, P-value<0.001), and MSWS-12 with TUG (r = 0.64, P-value<0.001) prior to the intervention. In addition, negative correlation coefficients were observed between BBS and MSIS-29 (r = -0.50,

**Table 1. Descriptive statistics of demographic and MS-related variables.**

| Variable | | Mean ± SD/Frequency (%) |
|---|---|---|
| **Baseline and demographic characteristics** | | |
| Age, years old[‡] | | 38.89 ± 9.85 |
| Sex | Female | 23 (76.7) |
| | Male | 7 (23.3) |
| PMH[†] | Negative | 23 (76.7) |
| | Thyroid disorders | 2 (6.7) |
| | Diabetes | 2 (6.7) |
| | Other | 3 (10.0) |
| Habitual history | Negative | 26 (86.7) |
| | Smoking | 4 (13.3) |
| | Alcohol | 1 (3.3) |
| | Hookah | 1 (3.3) |
| Education, years | ≤ 12 | 14 (46.7) |
| | > 12 | 16 (53.3) |
| Job status | Unemployed | 16 (53.3) |
| | Employed | 9 (30.0) |
| | Retired | 2 (6.7) |
| **MS-related characteristics** | | |
| MS phenotype | RRMS | 23 (76.7) |
| | SPMS | 4 (13.3) |
| | PPMS | 3 (10.0) |
| Age of disease onset, years old[§] | | 29.73 ± 8.82 |
| Disease duration, years[§] | | 8.57 ± 7.13 |
| EDSS score | | 4.2 ± 1.3 |
| DMD use | Not receiving DMD | 4 (13.3) |
| | Receiving DMD | 26 (86.7) |

Numeric data are described using mean ± SD. Categorial data are presented as numbers (%). Abbreviations: SD: Standard deviation, PMH: Past medical history, MS: Multiple sclerosis, EDSS: Expanded disability status scale, DMD: Disease-modifying drugs.

[†] Past medical history other than MS.

[‡] The patients' age at evaluation were obtained from their medical records, according to their birth certificate.

[§] The patients' age at MS diagnosis and the disease duration were obtained from in-person interviews.

P-value = 0.011), MSWS-12 (r = -0.44, P-value = 0.022), and TUG (r = -0.82, P-value<0.001). The Spearman correlation coefficients for other study time points are shown in **Fig 4**. **S6** and **S7 Tables in S1 File** present comprehensive details regarding the Spearman correlation coefficients along with confidence intervals for the total scores at each time point throughout the study.

## 4. Discussion

Our results demonstrated that daily caffeine consumption might enhance balance and functional mobility in PwMS (assessed by BBS and TUG). These improvements were noticed as soon as two weeks and persisted throughout the experiment (3 months). In addition, caffeine ingestion resulted in a considerable improvement in patients' reported MS-related QoL (assessed by MSIS-29) after two weeks. Despite improvement in the objective measurements of balance and functional mobility, as well as the subjective measurement of MS-related QoL,

**Table 2. The score trend of each criterion by times.**

| Criteria | Before | After 2 weeks | After 4 weeks | After 8 weeks | After 12 weeks | P |
|---|---|---|---|---|---|---|
| MSWS-12 score | 36.29 ± 13.55 | 31.47 ± 12.38 | 35.58 ± 11.29 | 36.46 ± 14.36 | 33.67 ± 13.92 | 0.583 |
| BBS score | 37.56 ± 13.03 | 42.77 ± 10.65 | 47.65 ± 8.09 | 46.42 ± 11.03 | 48.64 ± 11.60 | **<0.001** |
| TUG (second) | 31.25 ± 20.94 | 25.24 ± 16.20 | 23.88 ± 13.35 | 21.58 ± 15.75 | 19.30 ± 8.17 | **0.002** |
| MSIS-29 score | 84.38 ± 21.75 | 74.20 ± 24.60 | 73.95 ± 17.33 | 76.14 ± 23.45 | 76.00 ± 22.60 | **0.005** |
| PGIC score | N/A | 3.45 ± 1.68 | 3.72 ± 1.74 | 4.92 ± 1.51 | 4.10 ± 1.85 | 0.213 |

Mean ± standard deviation (SD) was reported for all variables. Abbreviations: P: Probability value, SD: Standard deviation, MSWS-12: 12-item Multiple Sclerosis Walking Scale, BBS: Berg Balance Scale, TUG: Timed Up-and-Go, MSIS-29: Multiple Sclerosis Impact Scale, PGIC: Patients' Global Impression of Change, NA: Not applicable.

the patient's self-reported ambulatory disability (evaluated by MSWS-12) and the patient's impression of overall change after the treatment (assessed by PGIC) did not considerably improve (**Fig 5**). To evaluate other factors associated with the outcome measures, we also examined the effect of sex and age on scores over time. Accordingly, being male was associated with a sharper increase in self-reported ambulatory disability (MSWS-12) over time than women. Additionally, while men tend to report lower perceived improvement in overall health

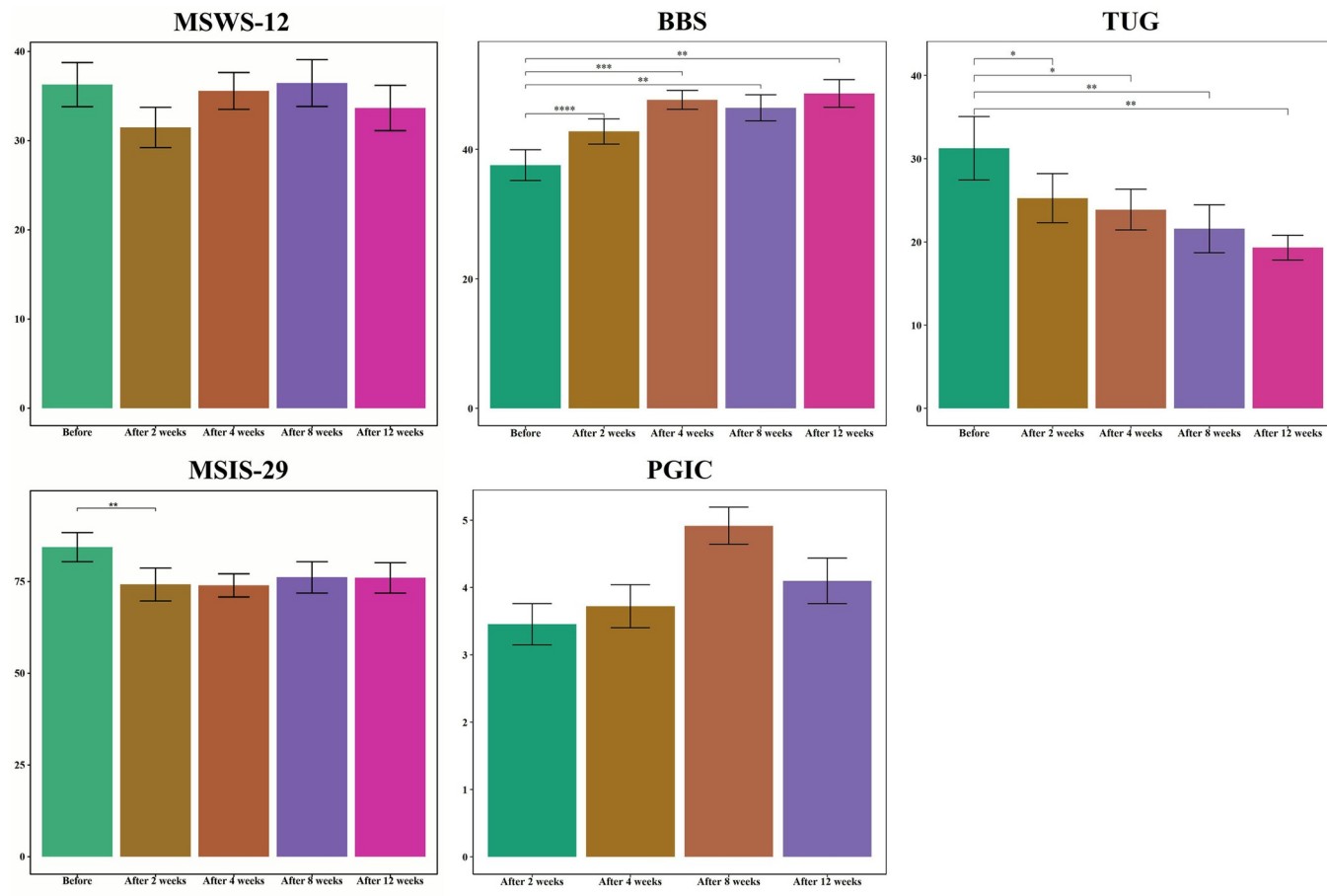

**Fig 3. The mean score of each criterion over the study period.** Abbreviations: MSWS-12: 12-item Multiple Sclerosis Walking Scale, BBS: Berg Balance Scale, TUG: Timed Up-and-Go, MSIS-29: Multiple Sclerosis Impact Scale, PGIC: Patients' Global Impression of Change.

**Table 3. The impact of sex on each criterion during the study period.**

| Criteria | | β (95% CI) | P-value |
|---|---|---|---|
| MSWS-12 | M vs. F | 3.00 (-6.47,12.64) | 0.543 |
| | Time | -4.12 (-6.25, -1.98) | <0.001 |
| | **(M vs. F) * Time** | **7.55 (2.95, 12.16)** | **0.001** |
| BBS | M vs. F | -8.79 (-18.90, 1.31) | 0.088 |
| | Time | 2.19 (0.97, 3.41) | <0.001 |
| | (M vs. F) * Time | -0.05 (-1.34, 1.24) | 0.940 |
| TUG | M vs. F | 13.43 (-12.35, 39.21) | 0.307 |
| | Time | -2.47 (-4.24, -0.69) | 0.006 |
| | (M vs. F) * Time | -1.87 (-5.81, 2.06) | 0.351 |
| MSIS-29 | M vs. F | 0.86 (-24.81, 26.52) | 0.948 |
| | Time | -4.21 (-7.37, -1.05) | 0.009 |
| | (M vs. F) * Time | -0.13 (-4.62, 4.36) | 0.955 |
| PGIC | M vs. F | -3.12 (-4.42, -1.83) | <0.001 |
| | Time | -0.08 (-0.24, 0.08) | 0.318 |
| | **(M vs. F) * Time** | **0.92 (0.66, 1.19)** | **<0.001** |

The generalized estimation equation (GEE) was used to evaluate the impact of sex on each criterion during the study period. Abbreviations: M: Male, F: Female, CI: Confidence interval, MSWS-12: 12-item Multiple Sclerosis Walking Scale, BBS: Berg Balance Scale, TUG: Timed Up-and-Go, MSIS-29: Multiple Sclerosis Impact Scale, PGIC: Patients' Global Impression of Change.

status (PGIC) at the end of the second week, the PGIC scores improved more prominently over time in men compared to women. Older age was associated with a worse balance status at baseline (lower BBS score). The impact of age on BBS scores increased as time progressed,

**Table 4. The impact of age on each criterion during the study period.**

| Variable | | β (95% CI) | P-value |
|---|---|---|---|
| MSWS-12 | Age | 0.32 (-0.15, 0.80) | 0.180 |
| | Time | -0.23 (-5.41, 4.95) | 0.930 |
| | Age * Time | 0.01 (-0.12, 0.13) | 0.924 |
| BBS | Age | -0.39 (-0.77, -0.02) | 0.039 |
| | Time | -1.89 (-4.45, 0.68) | 0.149 |
| | **Age * Time** | **0.09 (0.03, 0.15)** | **0.006** |
| TUG | Age | 0.42 (-0.48, 1.32) | 0.362 |
| | Time | 0.42 (-5.08, 5.92) | 0.882 |
| | Age * Time | -0.07 (-0.23, 0.10) | 0.419 |
| MSIS-29 | Age | 1.80 (0.88, 2.73) | <0.001 |
| | Time | -1.10 (-6.96, 4.76) | 0.712 |
| | Age * Time | -0.07 (-0.22, 0.09) | 0.404 |
| PGIC | Age | 0.03 (-0.06, 0.13) | 0.528 |
| | Time | 0.08 (-0.59, 0.74) | 0.825 |
| | Age * Time | 0.00 (-0.01, 0.02) | 0.891 |

The generalized estimation equation (GEE) was used to evaluate the impact of age on each criterion during the study period. Abbreviations: CI: Confidence interval, MSWS-12: 12-item Multiple Sclerosis Walking Scale, BBS: Berg Balance Scale, TUG: Timed Up-and-Go, MSIS-29: Multiple Sclerosis Impact Scale, PGIC: Patients' Global Impression of Change.

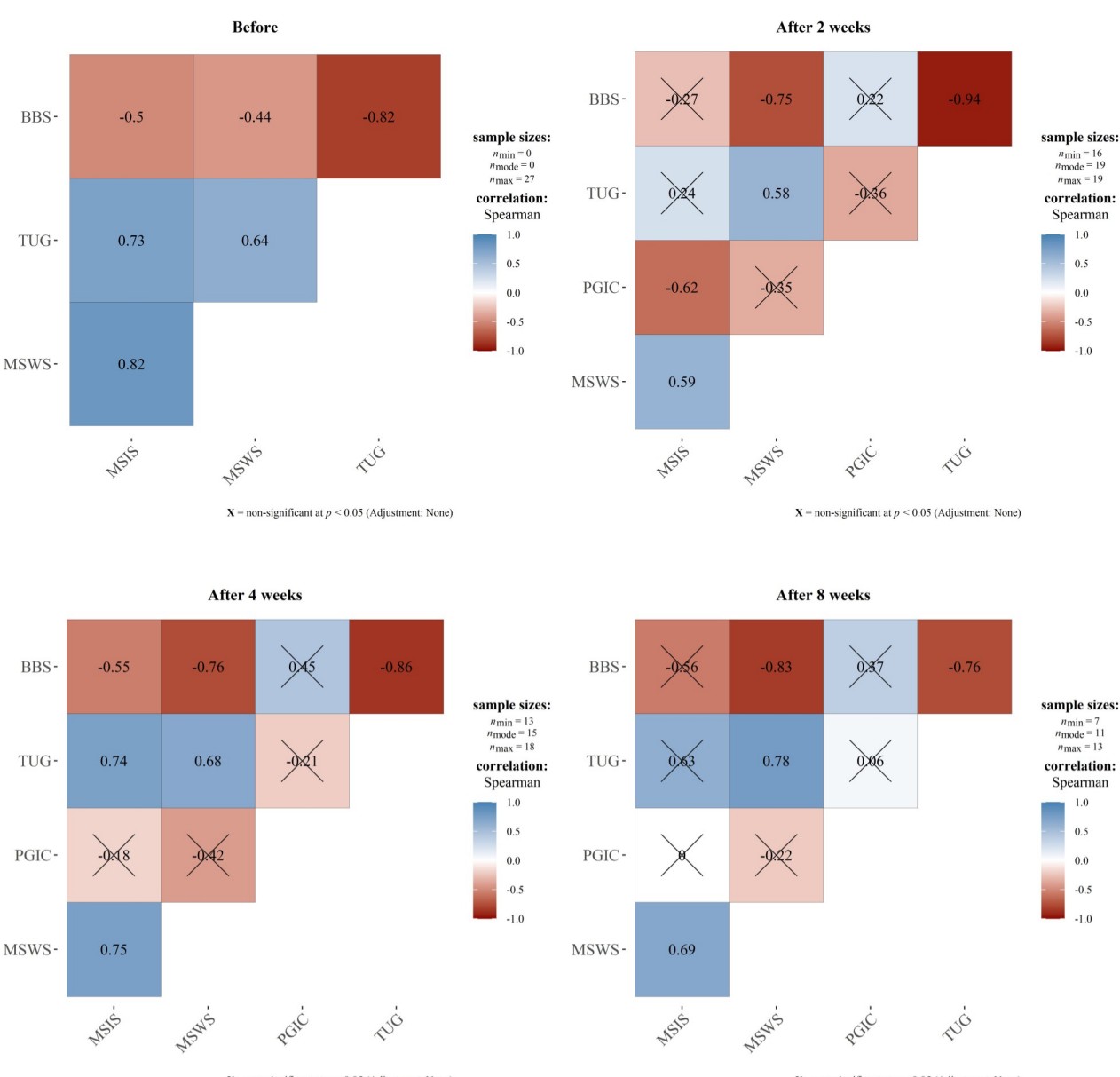

**Fig 4. The Spearman correlation coefficient between total scores.** Abbreviations: MSWS-12: 12-item Multiple Sclerosis Walking Scale, BBS: Berg Balance Scale, TUG: Timed Up-and-Go, MSIS-29: Multiple Sclerosis Impact Scale, PGIC: Patients' Global Impression of Change.

suggesting that the effects of aging on balance became more pronounced over longer periods. Finally, we evaluated the correlations between outcome measures, indicating meaningful correlations between MSWS-12, BBS, TUG, and MSIS-29. In other words, worse performance in each of these outcome measures was associated with a worse performance in all other three outcome measures, which is in line with expectations.

## 4.1 Preclinical and clinical evidence of caffeine's benefits in multiple sclerosis

According to an *in vivo* study on experimental autoimmune encephalomyelitis (EAE), an MS animal model, caffeine administration (0.2–2.0 mg/kg) reduced the neuroinflammatory

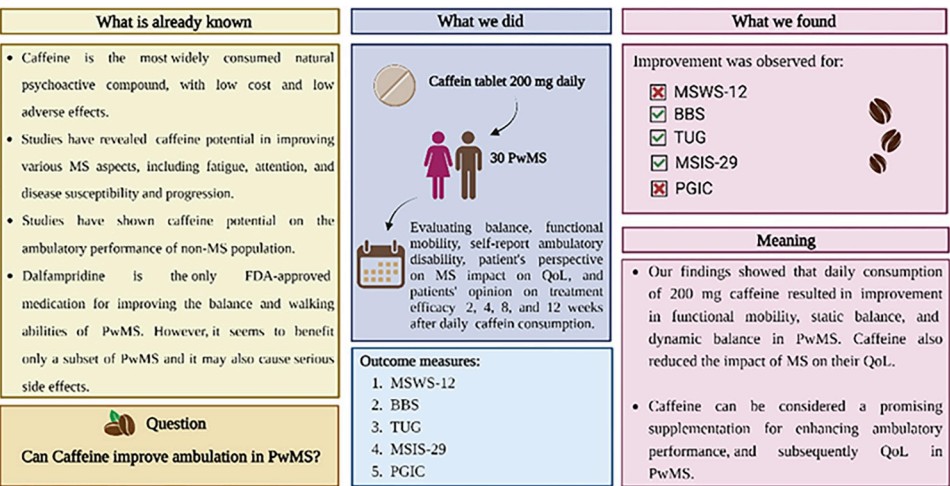

**Fig 5. Research summary.** Abbreviations: MSWS-12: 12-item Multiple Sclerosis Walking Scale, BBS: Berg Balance Scale, TUG: Timed Up-and-Go, MSIS-29: Multiple Sclerosis Impact Scale, PGIC: Patients' Global Impression of Change, QoL: Quality of life, PwMS: Persons with multiple sclerosis.

process and demyelination, through increasing the A1 adenosine receptor (A1AR) expression on macrophage/microglia [23]. Consistent with preclinical findings [23,24], clinical studies also have suggested possible beneficial effects of caffeine in various aspects of MS, including MS susceptibility, disease progression, reduced concentration, reduced attention, and cognitive dysfunction [18–20]. Reviewing the literature has suggested positive effects of caffeine on lowering the risk of MS susceptibility or ameliorating the clinical course of the disease [19]. Consistently, a large study on two population-representative case-control studies of PwMS found that high caffeine consumption (> 900 mL daily) has a protective role against MS development [21]. Likely, a cross-sectional study of 1372 PwMS discovered a possible positive effect of caffeine on the MS course and progression in patients with relapsing-remitting MS (RRMS), with a dose-effect relationship [22]. Another questionnaire-based study revealed that caffeine ingestion improved concentration, attention span, and performing structured daily routines in PwMS [18]. Consistently, a recent preliminary study suggested caffeine as a cognitive enhancer in PwMS, although this effect may be limited to tasks requiring a high level of attention [20].

## 4.2 Previous studies on caffeine's effects on balance and mobility in Non-MS populations

Although, according to our knowledge, there has not yet been a study evaluating caffeine effects on the balance and ambulatory performance of PwMS, previous studies have revealed caffeine's potential benefits on balance and some aspects of gait performance in populations other than PwMS [28–30,62,63]. Notably, the results of these studies considerably vary by the study population's age and sex. A recent randomized clinical trial in 30 young adults (20–35 years, female: 50%) indicated that caffeine ingestion (300–350 mg) resulted in improvements in postural balance and motor control [28]. Another study on 25 healthy middle-aged women (50–60 years) reported a positive effect of caffeine ingestion (100 mg) on postural balance during different situations (i.e., with closed eyes on the foam surface) [29]. Consistently, a trial on 20 middle-aged women (mean age: 52 years) recommended that caffeine consumption (100 mg) improved both cognitive and motor functions during static and dynamic dual-task

conditions [30]. On the other hand, a clinical trial of 12 healthy older adults (aged >65 years, female: 66.7%) found a negative effect of caffeine ingestion (3 mg/kg) on the bipedal standing balance and no effect on dynamic balance [62]. Another randomized clinical trial investigating the effects of taking caffeine (6 mg/kg) on the static balance and walking speed of 30 elderly patients (age ≥ 70 years, female: 50%) revealed that although caffeine improved postural stability with eyes open, it failed to improve walking speed [63]. However, it is worth mentioning that MS is frequently diagnosed in younger individuals [1] (our participants' mean age and mean age at MS diagnosis were nearly 39 and 30 years, respectively). Furthermore, available trials are limited by sample size, necessitating larger randomized clinical trials in this area.

## 4.3 Consideration of age and sex in balance and ambulatory performance studies

We also examined the effect of sex and age on outcome measures over the study period. In line with previous studies, our findings suggested the importance of taking age and sex into consideration while studying balance and ambulatory performance [28,62]. This is in corroboration with a systematic review, demonstrating that BBS scores worsen and become more unpredictable with age [64]. Additionally, our findings demonstrated that men reported more rapid deterioration of ambulatory abilities over time. Consistently, previous research has shown that sex hormones can influence disability progression, with men progressing earlier [65] and women progressing later during the perimenopausal period [66]. Notably, the male sex has been introduced as a negative prognostic factor of MS progression in several studies [65].

## 4.4 Limitations and strengths of the study

This study has several limitations, the most important of which are the relatively small sample size and a single-arm trial design. Although we designed a 2-week placebo run-in stage to reduce the placebo effect, lacking a control group limits the ability to determine whether the observed improvements were due to caffeine or to other factors, such as natural disease progression, placebo effect, or changes in other aspects of participants' lives during the study. Therefore, the results should be interpreted with caution, as they only suggest the "potential" efficacy of caffeine in improving balance and mobility in PwMS rather than a "confirmation of efficacy." However, single-arm trials are commonly used in phase II testing to collect preliminary evidence on potential treatment efficacy, obtain additional safety data, and assess whether a new treatment needs further investigation in a randomized phase III trial [67]. Another study limitation is attributed to the progressive increase in the number of dropouts. To address the issue of dropouts, we incorporated the GEE into our statistical analysis. The GEE is a robust method that accounts for missing data and provides unbiased estimates under the assumption of missing completely at random (MCAR) or missing at random (MAR) [68]. This approach enabled us to analyze the available data for each participant at different time points, accommodating the variability in the number of participants across assessments. Hence, despite the reduction in the number of participants over time, the GEE methodology empowered us to make valid inferences about the trends and changes observed in the outcome measures. Our study also lacked long-term follow-up (after 3 months) to evaluate the sustained effects of caffeine on balance and mobility in PwMS. Furthermore, our results may not be generalizable to all PwMS, as the potential effects of caffeine on patients with higher disability levels (EDSS ≥ 6) were not explored. Although, according to expert neurologists, no serious complications following caffeine consumption were observed in this study, the frequency of complications was not investigated. Another potential study limitation was that while we applied objective metrics to evaluate balance and functional mobility (BBS and TUG), MSWS-

12 provided a subjective assessment of the patients' walking capacity rather than an objective evaluation provided by tests such as the clinician-observed Timed 25-Foot Walk (T25FW). However, there is no clinician-administered scale evaluating all walking aspects in PwMS [69]. Additionally, while self-report questionnaires reflect the patients' performance during the day or week in their own environment, direct observation is usually performed over a brief period in the clinic and might not reflect patients' real-world experience [69]. MSWS-12 has also been shown to be more responsive than other walking-based measures, such as T25FW [69]. Notwithstanding these limitations, this is the first study so far documenting preliminary evidence of potential caffeine efficacy in improving balance and mobility in PwMS. This pilot study can serve as a foundation for more rigorous investigations, offering an initial estimate of the effect size to facilitate the planning of future controlled studies.

## 5. Conclusion and further direction

Although preliminary, our findings showed that daily consumption of 200 mg caffeine resulted in early and sustained improvement in functional mobility, static balance, and dynamic balance in PwMS. According to the patients, caffeine also reduced the impact of MS on their QoL. Hence, it can be considered a promising supplementation for enhancing balance and mobility and, subsequently, QoL in PwMS. The advantages of caffeine include low adverse effects, low cost, and the potential to improve other aspects of MS disease (such as cognitive aspects). Nevertheless, several questions remain to be answered. These results solely stem from this single-arm initial trial, which involved an Iranian population. They hint at the need for future definitive randomized placebo-controlled trials with larger sample sizes and longer follow-ups, involving diverse populations while taking age and sex into account. This approach is necessary to enhance our understanding of the efficacy of caffeine consumption and *confirm* its impact on the balance and mobility performance of PwMS. We suggest patients with higher disabilities also be included in future studies. We also suggest evaluating the effect of different caffeine doses on the balance and mobility of PwMS to determine the optimal treatment dosage. This would also be a fruitful area for further work to assess by affecting which MS-related symptoms (weakness, spasticity, fatigue, coordination alterations, cognitive dysfunction) can caffeine exert beneficial effects on patients' balance and mobility. It is also necessary to investigate the possible adverse effects of caffeine ingestion in PwMS, especially considering the long-term use of caffeine ($> 12$ weeks) and higher doses.

## Supporting information

**S1 File.** The supplementary material file includes Table S1: The CONSORT checklist of information to include when reporting a pilot trial, Table S2: The 12-item Multiple Sclerosis Walking Scale (MSWS-12), Table S3: The Berg Balance Scale (BBS), Figure S1: The Timed Up-and-Go (TUG), Table S4: The Multiple Sclerosis Impact Scale (MSIS-29), Table S5: The Patient's Global Impression of Changes (PGIC), and Figure S2: The impact of sex on each criterion during the study period. Supplementary Materials (containing the CONSORT checklist): Supplementary Materials
(DOCX)

**S2 File. IRCT registry, Original Language: Clinical Trial Protocol Registry (Persian).**
(PDF)

**S3 File. IRCT registry, English Language: Clinical Trial Protocol Registry (English).**
(PDF)

**S4 File. Study Protocol, English Language: Study Protocol (Persian).**
(PDF)

**S5 File. Study Protocol, English Language: Study Protocol (English).**
(PDF)

## Acknowledgments

We thank all the patients and clinicians who made this study possible.

## Author Contributions

**Conceptualization:** Afsoon Dadvar, Melika Jameie, Mohammadamin Parsaei, Mobina Amanollahi, Mahsa Babaei, Alireza Khosravi, Hamed Amirifard.

**Data curation:** Afsoon Dadvar, Mehdi Azizmohammad Looha.

**Formal analysis:** Mehdi Azizmohammad Looha, Meysam Zeynali Bujani.

**Investigation:** Afsoon Dadvar, Melika Jameie.

**Methodology:** Afsoon Dadvar, Melika Jameie, Alireza Khosravi, Hamed Amirifard.

**Project administration:** Alireza Khosravi, Hamed Amirifard.

**Resources:** Hamed Amirifard.

**Software:** Mehdi Azizmohammad Looha, Meysam Zeynali Bujani.

**Supervision:** Hamed Amirifard.

**Validation:** Mehdi Azizmohammad Looha.

**Visualization:** Melika Jameie, Hamed Amirifard.

**Writing – original draft:** Afsoon Dadvar, Melika Jameie, Mehdi Azizmohammad Looha, Mohammadamin Parsaei, Meysam Zeynali Bujani, Mobina Amanollahi, Mahsa Babaei.

**Writing – review & editing:** Melika Jameie, Alireza Khosravi, Hamed Amirifard.

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
