## [Decision Letter · Decision Letter 0]

4 Oct 2023

PONE-D-23-18917Efficacy of caffeine ingestion on the balance and gait in patients with multiple sclerosis: Preliminary evidence from a single-arm pilot clinical trialPLOS ONE

Dear Dr. Amirifard,

Thank you for submitting your manuscript to PLOS ONE. After careful consideration, we feel that the topic is of interest but does not fully meet PLOS ONE’s publication criteria as it currently stands. Therefore, we invite you to submit a revised version of the manuscript if you feel you are able to address the points raised during the review process.

Please submit any revised manuscript by Nov 18 2023 11:59PM. If you will need more time than this to complete your revisions, please reply to this message or contact the journal office at plosone@plos.org. Please include the following items when submitting your revised manuscript:A rebuttal letter that responds to each point raised by the academic editor and reviewer(s). You should upload this letter as a separate file labeled 'Response to Reviewers'.A marked-up copy of your manuscript that highlights changes made to the original version. You should upload this as a separate file labeled 'Revised Manuscript with Track Changes'.An unmarked version of your revised paper without tracked changes. You should upload this as a separate file labeled 'Manuscript'.

We look forward to receiving your revised manuscript.

Kind regards,

Antony Bayer

Academic Editor

PLOS ONE

2. We note that you have selected “Clinical Trial” as your article type. PLOS ONE requires that all clinical trials are registered in an appropriate registry (the WHO list of approved registries is at

"https://www.who.int/clinical-trials-registry-platform/network/primary-registries"" https://www.who.int/clinical-trials-registry-platform/network/primary-registries and more information on trial registration is at http://www.icmje.org/about-icmje/faqs/clinical-trials-registration/).

Please state the name of the registry and the registration number (e.g. ISRCTN or ClinicalTrials.gov) in the submission data and on the title page of your manuscript. a) Please provide the complete date range for participant recruitment and follow-up in the methods section of your manuscript. b) If you have not yet registered your trial in an appropriate registry, we now require you to do so and will need confirmation of the trial registry number before we can pass your paper to the next stage of review. Please include in the Methods section of your paper your reasons for not registering this study before enrolment of participants started. Please confirm that all related trials are registered by stating: “The authors confirm that all ongoing and related trials for this drug/intervention are registered”. Please see http://journals.plos.org/plosone/s/submission-guidelines#loc-clinical-trials for our policies on clinical trials.

5. We note that Figure S1 in your submission contain copyrighted images. All PLOS content is published under the Creative Commons Attribution License (CC BY 4.0), which means that the manuscript, images, and Supporting Information files will be freely available online, and any third party is permitted to access, download, copy, distribute, and use these materials in any way, even commercially, with proper attribution. For more information, see our copyright guidelines: http://journals.plos.org/plosone/s/licenses-and-copyright.

1. You may seek permission from the original copyright holder of Figure S1 to publish the content specifically under the CC BY 4.0 license.

6. We note that the original protocol that you have uploaded as a Supporting Information file contains an institutional logo. As this logo is likely copyrighted, we ask that you please remove it from this file and upload an updated version upon resubmission.

Additional Editor Comments:

1. The authors aim to “evaluated the effects on ambulatory performance and HRQoL” - how can effectiveness of an intervention be evaluated without any control?

2. The discussion should centre on the results and their implications and the background information from previous studies on hypothesis, benefits etc. should be in the Introduction.

3. The participants gave “fully informed consent” yet participants were “blind to the medication they received”? What were they told?

4.How was adherence assured (both for taking the study medication and for excluding other caffeinated products over the 14 weeks)?

5.The authors justify their sample size based on a reference to numbers suitable for a Phase 2 randomised trial (15 in each group, as originally planned in the placebo controlled study). How was this impacted by the eventual study just having a single arm (and by 18 not completing)? What effect size on the outcome measures would a sample size of 30 (and 12) have been able to demonstrate?

6.How did the analysis deal with the 18 dropouts? It seems surprising that the mean values and standard deviations for each assessment were broadly similar at each time point, despite progressively fewer people included.

7.What was the justification for choosing the 200mg dose of caffeine (and why is this different from the 2.5mg/Kg body weight) in the trial registry?

8.How were adverse events decided to be “treatment-related” and how did the 3 expert neurologists assess them?

9.What “deviations from the protocol” (p4) were there? Please explain all the differences from the protocol and the trial registry (age 20-55 included, weight >40Kg excluded, EDSS as primary outcome etc.)

10.What cut-off was used on trail-making task to exclude cognitive impairment?

11.For the PGIC, did a score of 4 indicate no change and so 1-3 a worsening, or did 1 to 7 indicate progressive improvement?

12.Mean age at MS diagnosis=29.73 and at evaluation=38.89 (difference 9.16 years), so how can mean disease duration be only 8.57 years?

Reviewers' comments:

Reviewer's Responses to Questions

**Comments to the Author**

1. Is the manuscript technically sound, and do the data support the conclusions?

Reviewer #1: Yes

Reviewer #2: Partly

2. Has the statistical analysis been performed appropriately and rigorously? 

Reviewer #1: Yes

Reviewer #2: No

3. Have the authors made all data underlying the findings in their manuscript fully available?

Reviewer #1: Yes

Reviewer #2: No

4. Is the manuscript presented in an intelligible fashion and written in standard English?

Reviewer #1: Yes

Reviewer #2: Yes

5. Review Comments to the Author

Reviewer #1: This study was aimed to evaluate its effects on ambulatory performance and health-related quality of life (HR-QoL) of patients with MS (PwMS). Authors concluded that Caffeine may enhance balance, functional mobility, and QoL in PwMS. Being male was associated with a sharper increase in self-reported ambulatory disability over time. The effects of aging on balance get more pronounced over time Overall, the study is interesting, however there are some clarifications needed.

Comment#1

Keywords: Please add quality of life to Keywords section.

Comment#2

Introduction, Line 87-90: Authors stated that this pilot single-arm phase II clinical trial set out to evaluate the potential effectiveness of caffeine ingestion on ambulatory performance (i.e., static and dynamic balance, functional mobility, and patient’s reported ambulatory disability) among PwMS. Ambulatory performance is not appropriate for static balance. Ambulatory performance is mostly related with dynamic balance rather than static balance.

Comment#3

Materials & Methods, Line 152. How do you assess the presence vestibular disorder or cognitive impairment?

Comment#4

Materials & Methods, Line 151-152. What cut-off point did you consider for the presence of moderate or severe anxiety assessed by The Hospital Anxiety and Depression Scale-Anxiety subscale?

Comment#5

Results, the authors should calculate effect size for each variable.

Reviewer #2: This manuscript presents data analysis from a non-randomized, Phase-II, pilot study on evaluating the effectiveness of caffeine ingestion on the balance and gait in MS patients. The topic is of importance, the study was registered as a RCT within the Iranian system, was approved by the respective IRB/Ethics Committee. While the study objectives sound interesting, is important, and on target, some shortcomings were observed, in regards to abiding by the CONSORT guidelines for conducting and reporting results of high-quality randomized controlled trials (RCTs). Some other (statistical) comments were also provided.

1. Methods:

Methods reporting need some work. An orderly manner is suggested, following CONSORT guidelines, without repeating information, such as Trial Design, Participant Eligibility Crtieria and settings, Interventions, Outcomes, sample size/power considerations, Interim analysis and stopping rules, etc. The authors are advised to create separate subsections for each of the possible topics (whichever necessary), and that way produce a very clear writeup. I see the Authors indeed made an attempt; however, they are advised to write it carefully, following nice examples in the manuscript below:

https://www.sciencedirect.com/science/article/pii/S0889540619300010

Specific comments:

(a) I am somewhat confused with the design! This is a single-arm, Phase-II, but I do not understand (justification not given clearly) behind the administration of placebo initially. Popular Phase-II designs, such as Simon's Phase-II, are often 2-staged. On the contrary, it would have been perfectly OK if a randomized design was considered (which often is much clear!). Why was that not conducted? Any water tight justification?

(b) Sample size/power: The sample size/power statement should reflect the statistical test used (one-sided/two-sided), the significance level (5%?), the corresponding effect size, etc. Even the trial is not randomized, one may compute using the "desired" change one wants to attain at the end of the study. Even pilot trials need to be conducted with some ballpark number. It should also be described in a separate sub-section.

(c) Statistical Analysis: Based on the (longitudinal) design of the study, the authors justifiably conducted a GEE analysis. Any thoughts, why a mixed linear model analysis was not conducted (I am not asking authors to do it)?

2. Results & Conclusions:

(a) The authors should check that any statement of significance should be followed by a p-value in the entire Results section. Otherwise, the Results section look OK.

(b) Conclusions should stress that findings are only based on this pilot trial (using an Iranian population), and allude to future larger trials/studies, combining other populations, to understand the effectiveness of caffeine intake.

6. PLOS authors have the option to publish the peer review history of their article (what does this mean?). If published, this will include your full peer review and any attached files.

Reviewer #1: **Yes: **Razieh Mofateh

Reviewer #2: No

---

## [Author Response · Author response to Decision Letter 0]

7 Dec 2023

* Note: Although the responses are also written here, please see the attached file (response to reviewers), as the figures and tables might not be displayed here correctly. 

Best wishes, 

PONE-D-23-18917

Efficacy of caffeine ingestion on the balance and gait in patients with multiple sclerosis: Preliminary evidence from a single-arm pilot clinical trial

PLOS ONE

Dear Dr. Amirifard,

Thank you for submitting your manuscript to PLOS ONE. After careful consideration, we feel that the topic is of interest but does not fully meet PLOS ONE’s publication criteria as it currently stands. Therefore, we invite you to submit a revised version of the manuscript if you feel you are able to address the points raised during the review process.

We look forward to receiving your revised manuscript.

Kind regards,

Antony Bayer

Academic Editor

PLOS ONE

Dear Professor Antony Bayer,

Thank you very much for providing us with the opportunity to strengthen our research. We sincerely appreciate all the precious comments from you and the respected reviewers. Having carefully considered the comments and suggestions, we have made all the relevant changes to our manuscript as outlined below in an itemized, point-by-point manner. We sincerely hope that these changes meet the approval criteria of the esteemed reviewers and the editorial board. 

Best Regards,

- Hamed Amirifard; MD, Assistant Professor of Neurology, Iranian Center of Neurological Research, Neuroscience Institute, Tehran University of Medical Sciences, Tehran, Iran. E-mail: Dr.amirifard@gmail.com, ORCID: 0000-0001-7675-5328, Phone: (+98) 912 6493820

- Melika Jameie; MD, MBA, Post-doctoral research fellow, Neuroscience Research Center, Iran University of Medical Sciences, Tehran, Iran; Iranian Center of Neurological Research, Neuroscience Institute, Tehran University of Medical Sciences, Tehran, Iran. ORCID: 0000-0002-2028-9935. Email: Jameiemelika@gmail.com; Ms-jameie@farabi.tums.ac.ir; Jameiemelika@sbmu.ac.ir

- Reply: Thank you very much. We tried our best to meet the journal’s style requirements. We sincerely hope that these changes meet the approval criteria of the PLOS ONE journal.

2. We note that you have selected “Clinical Trial” as your article type. PLOS ONE requires that all clinical trials are registered in an appropriate registry (the WHO list of approved registries). Please state the name of the registry and the registration number (e.g., ISRCTN or ClinicalTrials.gov) in the submission data and on the title page of your manuscript. 

- Reply: Thank you very much for your comment. This study was registered with the Iranian Registry of Clinical Trials (Registration number: IRCT2017012332142N1), a Primary Registry in the WHO Registry Network. Sincerely, we mentioned this in the Registration section. We also added the statement in the Title page, as your precious comment. 

a. Please provide the complete date range for participant recruitment and follow-up in the methods section of your manuscript.

- Reply: Thank you very much for your mention. According to your comment. in the Methods section, we mentioned this issue: “The first patient received treatment on March 9, 2017, and the trial ended on January 2, 2018.”

b. If you have not yet registered your trial in an appropriate registry, we now require you to do so and will need confirmation of the trial registry number before we can pass your paper to the next stage of review. Please include in the Methods section of your paper your reasons for not registering for this study before the enrolment of participants started. 

- Reply: Thank you for your mention. This study was registered with the Iranian Registry of Clinical Trials (Registration number: IRCT2017012332142N1), a Primary Registry in the WHO Registry Network. This information is available in the Registration section and the Title page.

c. Please confirm that all related trials are registered by stating: “The authors confirm that all ongoing and related trials for this drug/intervention are registered”. 

- Reply: Thank you very much. The authors confirm that all ongoing and related trials for this drug/intervention are registered with the Iranian Registry of Clinical Trials (Registration number: IRCT2017012332142N1).

3. We note that you have indicated that data from this study are available upon request. PLOS only allows data to be available upon request if there are legal or ethical restrictions on sharing data publicly. For information on unacceptable data access restrictions, please see http://journals.plos.org/plosone/s/data-availability#loc-unacceptable-data-access-restrictions. In your revised cover letter, please address the following prompts:

a. If there are ethical or legal restrictions on sharing a de-identified data set, please explain them in detail (e.g., data containing potentially identifying or sensitive patient information) and who has imposed them (e.g., an ethics committee). Please also provide contact information for a data access committee, ethics committee, or other institutional body to which data requests may be sent.

b. If there are no restrictions, please upload the minimal anonymized data set necessary to replicate your study findings as either Supporting Information files or to a stable, public repository and provide us with the relevant URLs, DOIs, or accession numbers. Please see http://www.bmj.com/content/340/bmj.c181.long for guidelines on how to de-identify and prepare clinical data for publication. For a list of acceptable repositories, please see http://journals.plos.org/plosone/s/data-availability#loc-recommended-repositories.

- Reply: Thank you very much for your comment. The ethics committee of the Zahedan University of Medical Sciences, Zahedan, Iran has imposed restrictions on sharing any dataset from this study. The university and the affiliated hospitals have strict policies that prohibit the sharing of such data to protect patient confidentiality and comply with legal and ethical regulations. Sincerely, for additional information or clarification regarding our data sharing policies and asking for the dataset, here is the contact information for our ethics committee: 

o Email: zaums.research@gmail.com

o The University Central Headquarters Call Center: (+98) 54 33372116

o Address: Zahedan University of Medical Sciences, Khalij -e- Fars Blvd, Zahedan, Sistan & Balouchestan Province, Islamic Republic of Iran.

o Website: https://enresearch.zaums.ac.ir/

We apologize for any inconvenience this may cause and appreciate your understanding of the legal and ethical constraints that prevent us from sharing the data. If you have any further questions or need additional information, please feel free to contact us or our ethics committee for further guidance. This information was also added in the “Revised cover letter”. 

4. PLOS requires an ORCID iD for the corresponding author in Editorial Manager on papers submitted after December 6th, 2016. Please ensure that you have an ORCID iD and that it is validated in Editorial Manager. To do this, go to ‘Update my Information’ (in the upper left-hand corner of the main menu), and click on the Fetch/Validate link next to the ORCID field. This will take you to the ORCID site and allow you to create a new ID or authenticate a pre-existing iD in Editorial Manager. Please see the following video for instructions on linking an ORCID iD to your Editorial Manager account: https://www.youtube.com/watch?v=_xcclfuvtxQ

- Reply: Thank you very much. The ORCID IDs for the two corresponding authors are provided in the Title page: 

o Hamed Amirifard, ORCID: 0000-0001-7675-5328

o Melika Jameie, ORCID: 0000-0002-2028-9935

5. We note that Figure S1 in your submission contains copyrighted images. All PLOS content is published under the Creative Commons Attribution License (CC BY 4.0), which means that the manuscript, images, and Supporting Information files will be freely available online, and any third party is permitted to access, download, copy, distribute, and use these materials in any way, even commercially, with proper attribution. For more information, see our copyright guidelines: http://journals.plos.org/plosone/s/licenses-and-copyright.

1. You may seek permission from the original copyright holder of Figure S1 to publish the content specifically under the CC BY 4.0 license. We recommend that you contact the original copyright holder with the Content Permission Form (http://journals.plos.org/plosone/s/file?id=7c09/content-permission-form.pdf) and the following text: “I request permission for the open-access journal PLOS ONE to publish XXX under the Creative Commons Attribution License (CCAL) CC BY 4.0 (http://creativecommons.org/licenses/by/4.0/). Please be aware that this license allows unrestricted use and distribution, even commercially, by third parties. Please reply and provide explicit written permission to publish XXX under a CC BY license and complete the attached form.” 

Please upload the completed Content Permission Form or other proof of granted permissions as an "Other" file with your submission. In the figure caption of the copyrighted figure, please include the following text: “Reprinted from [ref] under a CC BY license, with permission from [name of publisher], original copyright [original copyright year].”

2. If you are unable to obtain permission from the original copyright holder to publish these figures under the CC BY 4.0 license or if the copyright holder’s requirements are incompatible with the CC BY 4.0 license, please either i) remove the figure or ii) supply a replacement figure that complies with the CC BY 4.0 license. Please check the copyright information on all replacement figures and update the figure caption with source information. If applicable, please specify in the figure caption text when a figure is similar but not identical to the original image and is therefore for illustrative purposes only.

- Reply: Thank you very much for your consideration, and we sincerely apologize for inadvertently using this figure. We omitted the copyrighted figure and created a new figure. In the figure caption, we mentioned that “The figure is similar but not identical to the original image from TUG_test-print.pdf (cdc.gov) and is therefore for illustrative purposes only”. 

Figure S1. The figure is similar but not identical to the original image from TUG_test-print.pdf (cdc.gov) and is therefore for illustrative purposes only.

6. We note that the original protocol that you have uploaded as a Supporting Information file contains an institutional logo. As this logo is likely copyrighted, we ask that you please remove it from this file and upload an updated version upon resubmission.

- Reply: Thank you very much. We omitted the logo and the revised files are uploaded as “Revised Study Protocol, Original Language” and “Revised Study Protocol, English”. 

 

Additional Editor Comments:

1. The authors aim to “evaluate the effects on ambulatory performance and HRQoL” - how can the effectiveness of an intervention be evaluated without any control?

- Reply: Thank you very much for your meticulous comment. As with your precious comment and due to the fact that it was a pilot exploratory clinical trial with no control group, we revised all the “efficacy”-related terms to “potential efficacy” within the whole manuscript, including the Title, Abstract, and Introduction, reflecting the exploratory nature of our study. “Our trial's goal was to give an initial estimate of the effect size so that future controlled studies could be better planned”, even if it was not intended to prove caffeine's efficacy. This is especially crucial in the early phases of research when little is known about the possible effects of the intervention. 

- As per your valuable comment, in the Limitation section we also mentioned this important issue to acknowledge this study's limitation and the necessity to interpret the results cautiously: “Although we designed a 2-week placebo run-in stage to reduce the placebo effect, lacking a control group limits the ability to determine whether the observed improvements were due to caffeine or to other factors, such as natural disease progression, placebo effect, or changes in other aspects of participants' lives during the study”. 

- We also noted that: “Therefore, the results should be interpreted with caution, as they only suggest the “potential” efficacy of caffeine in improving the balance and mobility in PwMS rather than a “confirmation of efficacy.”

- Although this trial lacked a control group, it is worth noting that “single-arm trials are commonly used in phase II testing to collect preliminary evidence on potential treatment efficacy, obtain additional safety data, and assess whether a new treatment needs further investigation in a randomized phase III trial” (Clinical trial structures - PMC (nih.gov)). Therefore, by mentioning this point in the limitation section, we tried to aware the readers that this study only provides “preliminary evidence on potential treatment efficacy”. 

- In the Conclusion section, we also noted that “These results solely stem from this single-arm initial trial, which involved an Iranian population. They hint at the need for future definitive randomized placebo-controlled trials with larger sample sizes and longer follow-ups, involving diverse populations while taking age and sex into account. This approach is necessary to enhance our understanding of the efficacy of caffeine consumption and confirm its impact on balance and mobility performance of PwMS.” 

- At the end we again stress the need for further research to confirm the findings: “This pilot study can serve as a foundation for more rigorous investigations, offering an initial estimate of the effect size to facilitate the planning of future controlled studies”. 

- In summery, we revised the whole manuscript from Title to Conclusion to convey this concept to the readers that these findings do not “confirm” the efficacy, but rather suggest a “potential” efficacy. 

2. The discussion should center on the results and their implications and the background information from previous studies on hypothesis, benefits, etc. should be in the Introduction.

- Reply: Thank you very much for your valuable feedback. This pilot study suggested the possible potential benefits of caffeine on balance and mobility in PwMS. To our knowledge, this is the first study on this subject, limiting our ability to draw direct comparisons with existing literature. The scarcity of related articles compelled us to focus on two closely related areas: (1) studies related to the caffeine effect on other aspects of the lives of PwMS and (2) studies related to the caffeine effect on balance and mobility in populations other than PwMS. According to your valuable comment, we have streamlined the Discussion by integrating the subsection titled "The Hypothesis of Caffeine's Potential Benefits in Improving Balance and Mobility in PwMS" into the Introduction. The repeated parts were also omitted to increase readability. 

- Therefore, the study hypothesis and the reasons behind that are provided in the revised manuscript in the Introduction section: “Interventions, including physical rehabilitation (6), exercise, non-invasive brain stimulation (7, 8), and medications such as dalfampridine (4-aminopyridine) (9-11), nabiximols (12), polyunsaturated fatty acids, omega-3, omega-6 (13), and lipoic acid (14), have been evaluated for balance and gait improvement in PwMS (2, 6-15). Notably, dalfampridine is the only U.S. Food and Drug Administration (FDA)-approved medication for improving the balance and walking abilities of PwMS (Ampyra (dalfampridine) Information | FDA) (9-11, 15). However, it should be prescribed with caution, as it may cause serious side effects (16), including severe allergic reactions, seizures, and triggering/exacerbating trigeminal neuralgia (medication guide available at label (fda.gov)). Additionally, it seems that dalfampridine may help only a subset of PwMS, with one-quarter and one-third of patients experiencing faster walking speed and enhanced walking ability, respectively (16). 

Caffeine, a natural compound, is the most widely consumed psychoactive agent in the world (17). Studies have shown caffeine's potential benefits on various neurological disorders, including seizure, Alzheimer’s disease, Parkinson’s disease, stroke, and MS (18-25), possibly by reducing neuroinflammation and oxidative stress and increasing neurogenesis (26, 27). Specifically, research has highlighted the positive effects of caffeine on the ambulatory performance of non-MS populations (28-30), as well as on selected aspects of MS, including attention and disease progression (18-24). Aligned with caffeine's impact on the central nervous system, skeletal muscles (31), the ambulatory performance of non-MS populations (28-30), and specific aspects of MS (18-20, 23, 24), potentially favorable effects on balance and mobility in PwMS could also be anticipated. Consequently, we hypothesized that caffeine might have the potential to improve balance and mobility impairments as debilitating aspects of MS.”

3. The participants gave “fully informed consent” yet participants were “blind to the medication they received”? What were they told?

- Reply: Thank you very much for your valuable comment. The patients were told that they might receive either an active drug or a placebo during the study period. Indeed, they consented to receive either a placebo or medication; however, they did not know what they were receiving at each stage of the study. According to your precious comment, we revised the Methods> Blinding section and noted that “We described the nature of the study and the possibility of receiving either the active drug or a placebo without revealing the specific timing or details of when each would be administered. Participants were informed that at no point during the trial would they know which treatment they were getting, ensuring blinding during each stage, including both the 2-week pre-intervention placebo run-in stage and the 12-week post-intervention treatment stage”.

4. How was adherence assured (both for taking the study medication and for excluding other caffeinated products over the 14 weeks)?

- Reply: Thank you very much for your valuable comment. Since we had not mentioned this issue before, we revised the manuscript and provided the related information by adding a new section (Methods> Adherence monitoring): “Notably, throughout the course of the 14-week trial, medication compliance and avoidance of caffeinated products were carefully assessed, using several approaches (48, 49). First, before obtaining informed consent, patients were educated about the study's objectives, the significance of adhering to study requirements, the need to strictly follow the protocol, and detailed guidance on avoiding caffeinated products. Second, trained study personnel dispensed a certain amount of placebo or active medication during scheduled in-person visits (visits 1-6), ensuring that patients received only the required amount until their next visit. Additionally, at each visit, patients were reminded to bring the medication package from their previous visit and advised to abstain from caffeine-containing items. Third, paper-based patient diaries were employed to meticulously document all information related to medication intake, adverse events, and caffeine product consumption. Fourth, between visits, patients were contacted daily to confirm medication compliance, monitor potential adverse events, evaluate caffeine consumption, and address any other concerns. Eventually, participants were encouraged to reach out to the study team in case of difficulties, side effects affecting their compliance, or any violations involving the consumption of caffeinated products.”

5. The authors justify their sample size based on a reference to numbers suitable for a Phase 2 randomized trial (15 in each group, as originally planned in the placebo-controlled study). How was this impacted by the eventual study just having a single arm (and by 18 not completing)? What effect size on the outcome measures would a sample size of 30 (and 12) have been able to demonstrate?

- Reply: Thank you for your constructive input regarding the justification of our sample size, particularly in the context of the initially planned Phase 2 randomized trial and the subsequent decision to conduct a single-arm study, where 18 participants did not complete the study. We appreciate your consideration of the impact of these changes on our study design. In response to your query, the initial reference to a Phase 2 randomized trial with 15 participants in each group was made when planning a placebo-controlled study. Due to practical considerations (mentioned in the Methods > Trial design and any changes after trial commencement), we transitioned to a single-arm design during the course of the study, and unfortunately, 18 participants did not complete the trial. We acknowledge that this change has implications for the interpretation of our findings.

- Regarding the effect size, we employed GPower software with a conservative approach, selecting an effect size of 0.25 for the power analysis. This choice was intentionally lower than the actual observed effect size in our study. The resulting sample size calculation indicated a minimum of 21 for a power of 0.8 and 30 for a power of 0.95. Despite the conservative nature of these choices, we recognize the importance of providing a more detailed explanation of the impact of these decisions on our ability to detect meaningful differences in outcome measures.

- In light of your valuable feedback, we will further elaborate on the implications of our study design changes in the revised manuscript, addressing the specific effect size that a sample size of 30 (and 12, considering the 18 participants who did not complete the study) would have been able to demonstrate. We added this information in the sample size sub-section of the Methods, enhancing the transparency and completeness of our study design rationale: “The determination of sample size for this study involved the utilization of the GPower software. A deliberate underestimation was integrated, employing a conservative approach with an effect size established at 0.25, intentionally lower than the actual expected difference and effect size in our investigation. The analysis was conducted with a significance level of 0.05, and a range of power values from 0.8 to 0.95 was examined, incorporating considerations for five measurements and an assumed correlation of 0.5 between repeated measurements, which underestimated the true effect size and correlation observed in the study. Despite the conservative nature of these choices, the resulting sample size was computed as a minimum of 21 for a power of 0.8 and 30 for a power of 0.95.” We appreciate your guidance and remain committed to ensuring the thoroughness and clarity of our manuscript.

- The change in sample size can be elucidated as follows:

6. How did the analysis deal with the 18 dropouts? It seems surprising that the mean values and standard deviations for each assessment were broadly similar at each time point, despite progressively fewer people included.

- Reply: Thank you for your insightful observation regarding the analysis of dropouts in our study. We appreciate your attention to detail and the opportunity to clarify this aspect of our study. The apparent similarity in mean values and standard deviations for each assessment at different time points despite the progressive reduction in the number of participants is indeed a noteworthy point. We would like to assure you that we have considered this issue in our analysis.

- The handling of dropouts was addressed by utilizing the generalized estimation equation (GEE) in our statistical analysis. GEE is a robust method that accounts for missing data and provides unbiased estimates under the assumption of missing completely at random (MCAR) or missing at random (MAR) (1). In our case, the GEE approach allows us to analyze the available data for each participant at different time points, accommodating the variability in the number of participants across assessments.

- Despite the reduction in the number of participants over time, the GEE methodology enables us to make valid inferences about the trends and changes observed in the outcome measures. We acknowledge that the progressive decrease in the sample size is a limitation of our study, and we emphasized this point in the limitations section of the manuscript, as with your valuable comment: “Another study limitation is attributed to the progressive increase in the number of dropouts. To address the issue of dropouts, we incorporated the GEE into our statistical analysis. The GEE is a robust method that accounts for missing data and provides unbiased estimates under the assumption of missing completely at random (MCAR) or missing at random (MAR) (62). This approach enabled us to analyze the available data for each participant at different time points, accommodating the variability in the number of participants across assessments. Hence, despite the reduction in the number of participants over time, the GEE methodology empowered us to make valid inferences about the trends and changes observed in the outcome measures.”

7. What was the justification for choosing the 200 mg dose of caffeine (and why is this different from the 2.5mg/Kg body weight) in the trial registry?

- Reply: Thank you very much for your precious comment. According to your valuable comment, we revised the manuscript for two issues:

o Justification for deviation from the protocol: The choice of a fixed 200 mg/day dose of caffeine in our clinical trial was primarily driven by practical considerations. At the time of running the study, oral solutions of caffeine were not readily accessible in our country; hence, we decided to use accessible caffeine tablets to proceed with the trial. The use of caffeine tablets, rather than an oral solution, restricted our ability to administer a dose of 2.5 mg/kg. According to your valuable comment, this issue was added to the manuscript to assure transparency and describe any deviations from the protocol: “Furthermore, although the initial protocol was designed for a caffeine dosage of 2.5 mg/kg/day, we were compelled to utilize available caffeine tablets (200 mg/tablet) in the trial due to practical constraints and the unavailability of an oral caffeine solution”. 

o Justification for choosing a 200-mg dosage: Notably, 200 mg of caffeine is approximately equivalent to 2.5-3 mg/kg for a 60-70-kg adult. (https://www.sciencedirect.com/science/article/pii/S0149763416300690). Additionally, according to the European Food Safety Authority (EFSA), caffeine dosages up to 200 mg do not raise any safety concerns for non-pregnant adults (https://efsa.onlinelibrary.wiley.com/doi/abs/10.2903/j.efsa.2015.4102). Therefore, we mentioned that in the revised manuscript: “The selection of a 200 mg dosage was rationalized on two grounds: Firstly, 200 mg of caffeine corresponds to 2.5-3 mg/kg for an adult weighing between 60-70 kg. Secondly, according to the European Food Safety Authority (EFSA), caffeine doses of up to 200 mg do not elicit safety concerns in non-pregnant adults.”

8. How were adverse events decided to be “treatment-related” and how did the 3 expert neurologists assess them?

- Reply: Thank you very much for your valuable comment. For an adverse event to be considered treatment-related, we used the “NIA Adverse Event and Serious Adverse Event Guidelines” (https://www.nia.nih.gov/sites/default/files/2018-09/nia-ae-and-sae-guidelines-2018.pdf). According to this guideline, “to classify an adverse event as treatment-related, the neurologists took into account the logical temporal association with the treatment administration, the expected patterns of response, and the exclusion of other factors.” Sincerely, we added this information in the Methods> Safety monitoring section. 

9. What “deviations from the protocol” (p4) were there? Please explain all the differences between the protocol and the trial registry (age 20-55 included, weight > 40 kg excluded, EDSS as a primary outcome, etc.)

- Reply: Thank you very much for your valuable comments. We understand the importance of consistency between the study protocol and the trial registry, and we sincerely apologize for any inconvenience this may have caused.

o With regards to the weight criterion, patients weighing more than 40 kg were indeed included in the study. While this criterion was inadvertently forgotten to be written in the previous manuscript version, we have revised the manuscript to specify 'weight > 40 kg' as part of the inclusion criteria.

o In terms of the EDSS, although it was initially chosen as a primary outcome, we did not use it as such due to certain limitations in employing the EDSS as an outcome measure within a short duration, such as 3 months. First, while the EDSS is generally reliable for assessing patients over an extended duration (2), employing 3–6-month confirmed disability progression (sustained increase in EDSS) as a measure of disability might lead to an overestimation of permanent disability levels. This can be problematic for the interpretation of short-term trial results, potentially yielding misleading outcomes (3). Second, studies have suggested that the EDSS may not be as effective in consistently capturing subtle changes in disability for patients with mild or moderate disabilities, primarily due to its limited reliability, especially for lower EDSS scores (2). Additionally, the EDSS has demonstrated limited responsiveness in detecting clinically meaningful changes in disability and has shown limited predictive value in certain contexts (4). In summary, we did not use the EDSS as the primary outcome due to the challenges associated with its application, especially when attempting to measure short-term and minor changes in patients with lower levels of disability, as was the case in our study.

o Regarding the age criteria, our study adhered to an age range of 20-55, which was in accordance with the protocol (age range in our study: 23-55). We appreciate your attention to this discrepancy, and we apologize for any confusion it may have caused. We have since updated the manuscript to accurately reflect the age inclusion criterion. 

10. What cut-off was used on trail-making task to exclude cognitive impairment?

- Reply: Thank you very much for your precious comment. We used the Trail Making Test to evaluate cognitive impairment. TMT is a neuropsychological test that has been shown useful in predicting cognitive impairment (5). The TMT consists of two components: TMT-A and TMT-B. The task for each exam is for the individual to draw a line between consecutive circles that are set at random on a page. The TMT-A employs only numbers, while the TMT-B alternates between numbers and letters, forcing the patient to switch between them in a sequential sequence (5). Controversy exists about the best cut-off to determine impaired test results. Ideally, impairment in this test should be defined according to the normative data established based on age, education, ethnicity, and health status (6). Usually, impairment in each cognitive test is defined as “less than 1.5 z scores from the average of the normative population” (7). However, there is no normative data on the TMT test among the Iranian population, and the cutoff (TMT-B> 90 seconds) that we used was based on the previously published normative data (6, 8) and the clinical judgment of an MS specialist with more than 10 years experience. According to your valuable comment, the cut-off used to indicate significant cognitive impairment was added to the manuscript.

11. For the PGIC, did a score of 4 indicate no change and so 1-3 a worsening, or did 1 to 7 indicate progressive improvement?

- Reply: Thank you very much for your precious comment. We acknowledge that different versions of PGIC have been used in the literature. The one we used was similar to that in some previous studies, including https://www.ncbi.nlm.nih.gov/pmc/articles/PMC4623367/, Patients_Global_Impression_of_Change.pdf (chiro.org), https://www.health.mil/Reference-Center/Forms/2015/05/01/Patient-Global-Impression-Change-Scale, Assessing the clinical significance of change scores following carpal tunnel surgery - PMC (nih.gov), and Assessing the clinical significance of change scores recorded on subjective outcome measures - PubMed (nih.gov). In this scoring system, 1 represents “no change” and 7 represents “a great deal better”.

12. The mean age at MS diagnosis = 29.73 and at evaluation = 38.89 (difference 9.16 years), so how can the mean disease duration be only 8.57 years?

- Reply: Thank you very much for your meticulous feedback. The observed difference between the mean age at MS diagnosis (29.73 years) and the mean age at evaluation (38.89 years) compared to the reported mean disease duration (8.57 years) arises from differences in data collection methods. In our study, patients’ ages were obtained from their medical records, which were based on the date of birth recorded on their birth certificates. On the other hand, the age at disease onset and the disease duration were collected through patients’ interviews. While the difference (38.3 years old [29.73+8.57] vs. 38.9 years old) is relatively small (only six months), we fully acknowledge your concern and appreciate your attention to detail. We will consider this feedback for future studies and make efforts to minimize any recording disparities to enhance the uniformity of data collection. Your input is valuable in helping us improve the quality of our research.

- To increase transparency in reporting, we added this issue to the Table caption: “‡ The patients’ age at evaluation were obtained from their medical records, according to their birth certificate. §The patients’ age at MS diagnosis and the disease duration were obtained from in-person interviews.” 

Reviewers' comments:

Reviewer 1: 

This study aimed to evaluate its effects on ambulatory performance and health-related quality of life (HR-QoL) of patients with MS (PwMS). The authors concluded that Caffeine may enhance balance, functional mobility, and QoL in PwMS. Being male was associated with a sharper increase in self-reported ambulatory disability over time. The effects of aging on balance get more pronounced over time. Overall, the study is interesting, however, there are some clarifications needed.

- Reply: Dear reviewer, 

Thank you very much for your kind words and for providing us with the opportunity to strengthen our research. We sincerely appreciate your precious comments. Having carefully considered the comments and suggestions, we have made all the relevant changes to our manuscript as outlined below in an itemized, point-by-point manner. The revisions requested have been track-changed and highlighted within the manuscript in response.

Keywords: 

1. Please add quality of life to the Keywords section.

- Reply: Thank you very much for your comment. We added this keyword in the Keywords section. 

Introduction:

2. Line 87-90: Authors stated that this pilot single-arm phase II clinical trial set out to evaluate the potential effectiveness of caffeine ingestion on ambulatory performance (i.e., static and dynamic balance, functional mobility, and patient’s reported ambulatory disability) among PwMS. Ambulatory performance is not appropriate for static balance. Ambulatory performance is mostly related to dynamic balance rather than static balance.

- Reply: Thank you very much for your precious comment and for mentioning this issue. According to your valuable comment, we used the general term "Balance and Mobility" to address all of the outcome measures, including static balance, dynamic balance, ambulatory disability, functional mobility, and gait. We revised the whole manuscript from Title to Conclusion with respect to this issue and changed the term “ambulatory performance” to “balance and mobility”. 

Materials & Methods:

3. Line 152. How do you assess the presence of vestibular disorder or cognitive impairment?

- Reply: Thank you very much for your valuable comment. 

o For cognitive impairment: We used the Trail Making Test to evaluate cognitive impairment. TMT is a neuropsychological test that has been shown useful in predicting cognitive impairment (5). The TMT consists of two components: TMT-A and TMT-B. The task for each exam is for the individual to draw a line between consecutive circles that are set at random on a page. The TMT-A employs only numbers, while the TMT-B alternates between numbers and letters, forcing the patient to switch between them in a sequential sequence (5). Controversy exists about the best cut-off to determine impaired test results. Ideally, impairment in this test should be defined according to the normative data established based on age, education, ethnicity, and health status (6). Usually, impairment in each cognitive test is defined as “less than 1.5 z scores from the average of the normative population” (7). However, there is no normative data on the TMT test among the Iranian population, and the cutoff (TMT-B> 90 seconds) we used was based on the previously published normative data (6, 8) and the clinical judgment of an MS specialist with more than 10 years experience. According to your valuable comment, the definition used to indicate significant cognitive impairment was added to the manuscript. 

o For vestibular disorder: Vestibular disorders were ruled out of in the patient's examinations (cerebellar examinations, nystagmus examination, and head impulse test) no evidence was found in favor of peripheral vestibulopathy. According to your precious comment, we added this to the manuscript. 

4. Line 151-152. What cut-off point did you consider for the presence of moderate or severe anxiety assessed by The Hospital Anxiety and Depression Scale-Anxiety subscale?

- Reply: Thank you very much for your valuable comment. The Hospital Anxiety and Depression Scale-Anxiety subscale has a maximum score of 21. We used the Persian-validated translation of this scale (9). This scale is scored as below (10): 

o Scores < 7: non-cases

o Scores of 8-10: mild anxiety

o Scores of 11-14: moderate anxiety

o Scores of 15-21: severe anxiety

- Therefore score ≥ 11 was considered moderate to severe anxiety. According to your valuable comment, we added the cut-off in the revised manuscript. 

Results: 

5. The authors should calculate the effect size for each variable.

- Reply: Thank you very much for your valuable comment. We appreciate your careful review of our manuscript and your insightful comment regarding the calculation of effect sizes for each variable. We would like to clarify that the effect sizes in our study are interpreted differently based on the nature of the data presented in Tables 2, 3, and 4.

- In Table 2, the reported values represent mean differences (Before-After) for each criterion. These differences serve as indicators of the magnitude of change over time within each group, providing a measure of the effectiveness of the intervention.

- Conversely, Tables 3 and 4 present coefficients (β) derived from the generalized estimation equation (GEE) analysis, reflecting the impact of sex (Table 3) and age (Table 4) on the outcome measures during the study period. These coefficients can be considered analogous to effect sizes in the context of regression analysis, representing the estimated change in the criterion for a one-unit change in the predictor variable. Thank you again for your valuable feedback and please let us know if there is anything else we could provide. 

 

Reviewer 2: 

This manuscript presents data analysis from a non-randomized, Phase-II, pilot study on evaluating the effectiveness of caffeine ingestion on the balance and gait in MS patients. The topic is of importance, the study was registered as an RCT within the Iranian system and was approved by the respective IRB/Ethics Committee. While the study objectives sound interesting, are important, and are on target, some shortcomings were observed, in regard to abiding by the CONSORT guidelines for conducting and reporting results of high-quality randomized controlled trials (RCTs). Some other (statistical) comments were also provided.

- Reply: Dear reviewer, 

Thank you very much for your kind words and for providing us with the opportunity to strengthen our research. We sincerely appreciate your precious comments. Having carefully considered the comments and suggestions, we have made all the relevant changes to our manuscript as outlined below in an itemized, point-by-point manner. The revisions requested have been track-changed and highlighted within the manuscript in response.

Methods: 

1. Methods reporting needs some work. An orderly manner is suggested, following CONSORT guidelines, without repeating information, such as Trial Design, Participant Eligibility Criteria and settings, Interventions, Outcomes, sample size/power considerations, Interim analysis, stopping rules, etc. The authors are advised to create separate subsections for each of the possible topics (whichever is necessary), and that way produce a very clear writeup. I see the Authors indeed made an attempt; however, they are advised to write it carefully, following nice examples in the manuscript below: https://www.sciencedirect.com/science/article/pii/S0889540619300010

- Reply: Thank you very much for your precious comment. According to your valuable comment, we divided the method into the following subsections based on the CONSORT guideline and the study you provided: 

o Trial design and any changes after trial commencement

o Participants 

o Study settings and Ethics statement (sincerely, although this section could be merged with the “Participants” section as per the CONSORT guideline as well as the article you mentioned, it has been divided into two subsections owing to the length of the text.) 

o Intervention (this section is newly added). 

o Outcomes

o Study measures (Although not explicitly outlined in the CONSORT guidelines, this section is included to offer detailed information on outcome measures that may not be familiar to all readers of the journal.)

o Sample size 

o Randomization, sequence generation, allocation concealment mechanism, implementation: not applicable

o Blinding

o Adherence monitoring (Although not explicitly outlined in the CONSORT guidelines, this section is included to offer detailed information on how we assessed patients’ medication compliance and avoidance of caffeinated products.) 

o Safety monitoring (While not explicitly specified in the CONSORT guidelines, this section is incorporated to provide comprehensive details on how our safety monitoring team evaluated patients and ensured the identification of any treatment-related adverse events.)

o Statistical analysis 

- We omitted the repeated information according to your valuable comment. Additionally, based on the updates in the revised manuscript, the CONSORT checklist (Table S1) was also updated. 

2. I am somewhat confused with the design! This is a single-arm, Phase II, but I do not understand (justification not given clearly) behind the administration of placebo initially. Popular Phase-II designs, such as Simon's Phase-II, are often 2-staged. On the contrary, it would have been perfectly OK if a randomized design was considered (which often is much clearer!). Why was that not conducted? Any watertight justification?

- Reply: Thank you for your valuable comment, and we sincerely apologize for any confusion. The decision to transition from a double-armed design to a single-armed study with a placebo run-in was driven by financial constraints encountered during the study. The initial plan involved two arms, with one group receiving caffeine and the other a placebo. However, the projected cost of 2520 placebos (30 patients*84 days) proved financially unfeasible for our university, especially given Zahedan's status as one of the low-income cities in Iran. To address this challenge while mitigating potential bias, we opted for a single-armed trial with a two-week placebo run-in. This decision aimed to balance financial constraints and the need to account for the placebo effect. While not exactly the same, the placebo run-in approach has been utilized by other researchers (e.g., Sustained-release oral fampridine in multiple sclerosis: a randomised, double-blind, controlled trial - PubMed (nih.gov)). In this two-week placebo run-in, only 420 placebos (30 patients*14 days) were required. 

- We acknowledge this study's limitation and, in the interest of transparency, have explained the change in the Methods section: “Notably, the initial study protocol involved a double-armed design, with one group receiving caffeine and another group receiving a placebo. However, due to financial constraints encountered during the study, it became necessary to modify the protocol and proceed as a single-armed study. To mitigate potential bias, a two-week placebo run-in stage was incorporated before the investigation.” Furthermore, in the limitations section, we explicitly stated “Although we designed a 2-week placebo run-in stage to reduce the placebo effect, lacking a control group limits the ability to determine whether the observed improvements were due to caffeine or to other factors, such as natural disease progression, placebo effect, or changes in other aspects of participants' lives during the study. Therefore, the results should be interpreted with caution, as they only suggest the “potential” efficacy of caffeine in improving balance and mobility in PwMS rather than a “confirmation of efficacy.”

- Furthermore, recognizing that the study design may be confusing to readers, we have visualized the design in Figure 2 to enhance comprehension.

- Thank you again and please let us know if we can provide any other explanation to increase the study transparency. 

3. Sample size/power: The sample size/power statement should reflect the statistical test used (one-sided/two-sided), the significance level (5%?), the corresponding effect size, etc. Even if the trial is not randomized, one may compute using the "desired" change one wants to attain at the end of the study. Even pilot trials need to be conducted with some ballpark number. It should also be described in a separate sub-section.

- Reply: Thank you for your thoughtful and constructive comments on our manuscript. We appreciate your thorough review and have carefully considered your suggestions regarding the sample size and power analysis.

- In response to your comment, we used GPower software to determine the sample size for our study. We employed a conservative approach by selecting an effect size of 0.25, which is lower than the actual difference and effect size observed in our study. Additionally, we set a significance level of 0.05 and explored a range of power values from 0.8 to 0.95. Our analysis included five measurements and assumed a correlation of 0.5 between repeated measurements, which is lower than the actual correlation observed in our study. Despite these conservative choices, the calculated sample size was found to be a minimum of 21 for a power of 0.8 and 30 for a power of 0.95. It's important to note that our effect size and correlation values in the actual study are higher than those used in the power analysis, which would, in turn, require a smaller sample size for equivalent statistical power.

- The change in sample size can be elucidated as follows:

- We understand the importance of providing a comprehensive description of the sample size determination process, including details about the statistical test used, the significance level, and the effect size. Hence, the following paragraph was added to the sample size section of the Methods: “The determination of sample size for this study involved the utilization of the GPower software. A deliberate underestimation was integrated, employing a conservative approach with an effect size established at 0.25, intentionally lower than the actual expected difference and effect size in our investigation. The analysis was conducted with a significance level of 0.05, and a range of power values from 0.8 to 0.95 was examined, incorporating considerations for five measurements and an assumed correlation of 0.5 between repeated measurements, which underestimated the true effect size and correlation observed in the study. Despite the conservative nature of these choices, the resulting sample size was computed as a minimum of 21 for a power of 0.8 and 30 for a power of 0.95.”

4. Statistical Analysis: Based on the (longitudinal) design of the study, the authors justifiably conducted a GEE analysis. Any thoughts, on why a mixed linear model analysis was not conducted (I am not asking authors to do it)?

- Reply: Thank you for your comment regarding the statistical analysis methods employed in our study. We appreciate the opportunity to address your question regarding the choice of the generalized estimation equation (GEE) over a mixed linear model (MLM) analysis in our longitudinal study design. The decision to use GEE instead of MLM was carefully considered based on the following factors:

o Distributional Assumptions: GEE is a robust method for analyzing longitudinal data, especially when the distributional assumptions of MLM, such as normality and homoscedasticity, may not be met. GEE makes fewer assumptions about the distribution of the outcome variables, making it more suitable for non-normally distributed data or when the variance may be non-constant across time points (In our study, some of variables were not normal)

o Population-Averaged Estimates: GEE provides population-averaged estimates, which are often more relevant in epidemiological and clinical research where the focus is on the average response across the population rather than individual-specific changes.

o Model Interpretability: GEE allows for a straightforward interpretation of the regression coefficients, especially when focusing on population-averaged effects. This can enhance the practical significance and communication of the results in the context of our study.

- While MLM could also be a valid approach, the above considerations led us to choose GEE as the more appropriate method for our specific research context.

Results & Conclusions:

5. The authors should check that any statement of significance should be followed by a p-value in the entire Results section. Otherwise, the Results section looks OK.

- Reply: Thank you very much for your valuable comment. In response to your comment, we have carefully examined the Results section and ensured that each statement of significance is accompanied by the corresponding p-value.: 

o “Accordingly, PGIC was significantly lower for males than for females at the end of the second week (β: -3.12, 95% CI: -4.42, -1.83, P-value<0.001).”

- We also present the updated Results section with the inclusion of p-values for the Spearman correlation coefficients, as with your precious comment: “Eventually, we assessed the correlations between outcome measures, including the MSWS-12, BBS, TUG, MSIS-29, and PGIC (Figure 4). The results showed significant positive correlations between MSIS-29 and MSWS-12 (r=0.82, p<0.001), MSIS-29 with TUG (r=0.73, p<0.001), and MSWS-12 with TUG (r=0.64, p<0.001) prior to the intervention. In addition, negative correlation coefficients were observed between BBS and MSIS-29 (r=-0.50, p=0.011), MSWS-12 (r=-0.44, p=0.022), and TUG (r=-0.82, p<0.001). The Spearman correlation coefficients for other study time points are shown in Figure 4.”

- Additionally, we added the following tables regarding the “Spearman correlation coefficients along with confidence intervals for the total scores at each timepoint throughout the study” to the supplementary materials (Tables S6 and S7): “Tables S6 and S7 present comprehensive details regarding the Spearman correlation coefficients along with confidence intervals for the total scores at each timepoint throughout the study.”

Correlations

Time BBS TUG PGIC MSWS MSIS

Before Spearman's rho BBS Correlation Coefficient 1.000 -.820** . -.440* -.498*

 Sig. (2-tailed) . .000 . .022 .011

 N 27 25 0 27 25

 TUG Correlation Coefficient -.820** 1.000 . .642** .732**

 Sig. (2-tailed) .000 . . .000 .000

 N 25 26 0 26 24

 PGIC Correlation Coefficient . . . . .

 Sig. (2-tailed) . . . . .

 N 0 0 0 0 0

 MSWS Correlation Coefficient -.440* .642** . 1.000 .822**

 Sig. (2-tailed) .022 .000 . . .000

 N 27 26 0 28 26

 MSIS Correlation Coefficient -.498* .732** . .822** 1.000

 Sig. (2-tailed) .011 .000 . .000 .

 N 25 24 0 26 26

After 2 weeks Spearman's rho BBS Correlation Coefficient 1.000 -.944** .219 -.748** -.265

 Sig. (2-tailed) . .000 .368 .000 .287

 N 22 16 19 18 18

 TUG Correlation Coefficient -.944** 1.000 -.360 .576* .241

 Sig. (2-tailed) .000 . .155 .016 .352

 N 16 17 17 17 17

 PGIC Correlation Coefficient .219 -.360 1.000 -.353 -.620**

 Sig. (2-tailed) .368 .155 . .151 .005

 N 19 17 22 18 19

 MSWS Correlation Coefficient -.748** .576* -.353 1.000 .595**

 Sig. (2-tailed) .000 .016 .151 . .007

 N 18 17 18 19 19

 MSIS Correlation Coefficient -.265 .241 -.620** .595** 1.000

 Sig. (2-tailed) .287 .352 .005 .007 .

 N 18 17 19 19 20

After 4 weeks Spearman's rho BBS Correlation Coefficient 1.000 -.860** .448 -.757** -.551*

 Sig. (2-tailed) . .000 .082 .001 .022

 N 17 13 16 16 17

 TUG Correlation Coefficient -.860** 1.000 -.210 .682** .744**

 Sig. (2-tailed) .000 . .452 .005 .001

 N 13 15 15 15 15

 PGIC Correlation Coefficient .448 -.210 1.000 -.416 -.181

 Sig. (2-tailed) .082 .452 . .097 .473

 N 16 15 18 17 18

 MSWS Correlation Coefficient -.757** .682** -.416 1.000 .746**

 Sig. (2-tailed) .001 .005 .097 . .000

 N 16 15 17 19 18

 MSIS Correlation Coefficient -.551* .744** -.181 .746** 1.000

 Sig. (2-tailed) .022 .001 .473 .000 .

 N 17 15 18 18 19

After 8 weeks Spearman's rho BBS Correlation Coefficient 1.000 -.757* .372 -.826** -.557

 Sig. (2-tailed) . .049 .324 .002 .060

 N 12 7 9 11 12

 TUG Correlation Coefficient -.757* 1.000 .057 .778* .633

 Sig. (2-tailed) .049 . .875 .014 .067

 N 7 11 10 9 9

 PGIC Correlation Coefficient .372 .057 1.000 -.215 -.005

 Sig. (2-tailed) .324 .875 . .525 .989

 N 9 10 12 11 11

 MSWS Correlation Coefficient -.826** .778* -.215 1.000 .691**

 Sig. (2-tailed) .002 .014 .525 . .009

 N 11 9 11 13 13

 MSIS Correlation Coefficient -.557 .633 -.005 .691** 1.000

 Sig. (2-tailed) .060 .067 .989 .009 .

 N 12 9 11 13 14

After 12 weeks Spearman's rho BBS Correlation Coefficient 1.000 -.690* -.099 -.478 -.185

 Sig. (2-tailed) . .040 .800 .162 .610

 N 11 9 9 10 10

 TUG Correlation Coefficient -.690* 1.000 .017 .610 .600

 Sig. (2-tailed) .040 . .965 .081 .088

 N 9 11 9 9 9

 PGIC Correlation Coefficient -.099 .017 1.000 -.512 -.641

 Sig. (2-tailed) .800 .965 . .159 .063

 N 9 9 10 9 9

 MSWS Correlation Coefficient -.478 .610 -.512 1.000 .781**

 Sig. (2-tailed) .162 .081 .159 . .005

 N 10 9 9 12 11

 MSIS Correlation Coefficient -.185 .600 -.641 .781** 1.000

 Sig. (2-tailed) .610 .088 .063 .005 .

 N 10 9 9 11 11

**. Correlation is significant at the 0.01 level (2-tailed).

*. Correlation is significant at the 0.05 level (2-tailed).

Confidence Intervals of Spearman's rho

Time Spearman's rho Significance(2-tailed) 95% Confidence Intervals (2-tailed)a,b

 Lower Upper

Before BBS - TUG -.820 .000 -.920 -.621

 BBS - PGIC .c . . .

 BBS - MSWS -.440 .022 -.708 -.060

 BBS - MSIS -.498 .011 -.752 -.116

 TUG - PGIC .c . . .

 TUG - MSWS .642 .000 .328 .828

 TUG - MSIS .732 .000 .456 .879

 PGIC - MSWS .c . . .

 PGIC - MSIS .c . . .

 MSWS - MSIS .822 .000 .630 .919

After 2 weeks BBS - TUG -.944 .000 -.981 -.838

 BBS - PGIC .219 .368 -.275 .621

 BBS - MSWS -.748 .000 -.903 -.419

 BBS - MSIS -.265 .287 -.660 .244

 TUG - PGIC -.360 .155 -.724 .160

 TUG - MSWS .576 .016 .116 .832

 TUG - MSIS .241 .352 -.286 .656

 PGIC - MSWS -.353 .151 -.711 .151

 PGIC - MSIS -.620 .005 -.843 -.217

 MSWS - MSIS .595 .007 .179 .830

After 4 weeks BBS - TUG -.860 .000 -.959 -.574

 BBS - PGIC .448 .082 -.077 .779

 BBS - MSWS -.757 .001 -.913 -.404

 BBS - MSIS -.551 .022 -.821 -.080

 TUG - PGIC -.210 .452 -.662 .353

 TUG - MSWS .682 .005 .246 .889

 TUG - MSIS .744 .001 .359 .912

 PGIC - MSWS -.416 .097 -.754 .097

 PGIC - MSIS -.181 .473 -.607 .326

 MSWS - MSIS .746 .000 .416 .902

After 8 weeks BBS - TUG -.757 .049 -.964 .020

 BBS - PGIC .372 .324 -.407 .838

 BBS - MSWS -.826 .002 -.955 -.433

 BBS - MSIS -.557 .060 -.862 .044

 TUG - PGIC .057 .875 -.608 .675

 TUG - MSWS .778 .014 .214 .953

 TUG - MSIS .633 .067 -.077 .917

 PGIC - MSWS -.215 .525 -.732 .458

 PGIC - MSIS -.005 .989 -.616 .610

 MSWS - MSIS .691 .009 .209 .903

After 12 weeks BBS - TUG -.690 .040 -.932 -.023

 BBS - PGIC -.099 .800 -.727 .620

 BBS - MSWS -.478 .162 -.857 .237

 BBS - MSIS -.185 .610 -.740 .520

 TUG - PGIC .017 .965 -.668 .686

 TUG - MSWS .610 .081 -.114 .911

 TUG - MSIS .600 .088 -.130 .908

 PGIC - MSWS -.512 .159 -.883 .253

 PGIC - MSIS -.641 .063 -.919 .063

 MSWS - MSIS .781 .005 .323 .943

a. Estimation is based on Fisher's r-to-z transformation.

b. Estimation of standard error is based on the formula proposed by Fieller, Hartley, and Pearson.

c. Cannot be computed because at least one of the variables is constant.

Thank you again for guiding us to enhance the quality of our manuscript.

6. Conclusions should stress that findings are only based on this pilot trial (using an Iranian population), and allude to future larger trials/studies, combining other populations, to understand the effectiveness of caffeine intake.

- Reply: Thank you very much for your precious comment. According to your valuable comment in the Conclusion section, we emphasized that: “These results solely stem from this single-arm initial trial, which involved an Iranian population. They hint at the need for future definitive randomized placebo-controlled trials with larger sample sizes and longer follow-ups, involving diverse populations while taking age and sex into account. This approach is necessary to enhance our understanding of the efficacy of caffeine consumption and confirm its impact on balance and mobility performance of PwMS.”

 

References:

1. Fitzmaurice GM, Laird NM. Generalized linear mixture models for handling nonignorable dropouts in longitudinal studies. Biostatistics. 2000;1(2):141-56.

2. Meyer-Moock S, Feng YS, Maeurer M, Dippel FW, Kohlmann T. Systematic literature review and validity evaluation of the Expanded Disability Status Scale (EDSS) and the Multiple Sclerosis Functional Composite (MSFC) in patients with multiple sclerosis. BMC Neurol. 2014;14:58.

3. Kalincik T, Cutter G, Spelman T, Jokubaitis V, Havrdova E, Horakova D, et al. Defining reliable disability outcomes in multiple sclerosis. Brain. 2015;138(11):3287-98.

4. Kragt JJ, Thompson AJ, Montalban X, Tintoré M, Río J, Polman CH, et al. Responsiveness and predictive value of EDSS and MSFC in primary progressive MS. Neurology. 2008;70(13 Pt 2):1084-91.

5. Stebbins GT. Chapter 27 - Neuropsychological Testing. In: Goetz CG, editor. Textbook of Clinical Neurology (Third Edition). Philadelphia: W.B. Saunders; 2007. p. 539-57.

6. Tombaugh TN. Trail Making Test A and B: normative data stratified by age and education. Archives of clinical neuropsychology. 2004;19(2):203-14.

7. Eshaghi A, Riyahi-Alam S, Roostaei T, Haeri G, Aghsaei A, Aidi MR, et al. Validity and reliability of a Persian translation of the Minimal Assessment of Cognitive Function in Multiple Sclerosis (MACFIMS). The Clinical neuropsychologist. 2012;26(6):975-84.

8. Giovagnoli AR, Del Pesce M, Mascheroni S, Simoncelli M, Laiacona M, Capitani E. Trail making test: normative values from 287 normal adult controls. Ital J Neurol Sci. 1996;17(4):305-9.

9. Montazeri A, Vahdaninia M, Ebrahimi M, Jarvandi S. The Hospital Anxiety and Depression Scale (HADS): translation and validation study of the Iranian version. Health Qual Life Outcomes. 2003;1:14.

10. Stern AF. The Hospital Anxiety and Depression Scale. Occupational Medicine. 2014;64(5):393-4.

---

## [Decision Letter · Decision Letter 1]

2 Jan 2024

Potential efficacy of caffeine ingestion on balance and mobility in patients with multiple sclerosis: Preliminary evidence from a single-arm pilot clinical trial

PONE-D-23-18917R1

Dear Dr. Amirifard,

Thank you for your detailed and considered responses to the review comments following your initial submission of this paper. We’re pleased to inform you that your revised manuscript has been judged scientifically suitable for publication and will be formally accepted for publication once it meets all outstanding technical requirements.

Kind regards,

Antony Bayer

Academic Editor

PLOS ONE

Additional Editor Comments (optional):

Reviewers' comments:

Reviewer's Responses to Questions

**Comments to the Author**

1. If the authors have adequately addressed your comments raised in a previous round of review and you feel that this manuscript is now acceptable for publication, you may indicate that here to bypass the “Comments to the Author” section, enter your conflict of interest statement in the “Confidential to Editor” section, and submit your "Accept" recommendation.

Reviewer #1: All comments have been addressed

Reviewer #2: All comments have been addressed

2. Is the manuscript technically sound, and do the data support the conclusions?

Reviewer #1: Yes

Reviewer #2: (No Response)

3. Has the statistical analysis been performed appropriately and rigorously? 

Reviewer #1: Yes

Reviewer #2: (No Response)

4. Have the authors made all data underlying the findings in their manuscript fully available?

Reviewer #1: Yes

Reviewer #2: (No Response)

5. Is the manuscript presented in an intelligible fashion and written in standard English?

Reviewer #1: Yes

Reviewer #2: (No Response)

6. Review Comments to the Author

Reviewer #1: (No Response)

Reviewer #2: (No Response)

7. PLOS authors have the option to publish the peer review history of their article (what does this mean?). If published, this will include your full peer review and any attached files.

Reviewer #1: No

Reviewer #2: No

---

## [Editor Report · Acceptance letter]

2 Feb 2024

PONE-D-23-18917R1 

PLOS ONE

Dear Dr. Amirifard, 

I'm pleased to inform you that your manuscript has been deemed suitable for publication in PLOS ONE. Congratulations! Your manuscript is now being handed over to our production team.

Kind regards, 

on behalf of

Professor Antony Bayer 

Academic Editor

PLOS ONE